# Post-transcriptional suppression of the pioneer factor Zelda protects the adult *Drosophila* testis from activation of the ovary program

Sneh Harsh[1], Hsiao-Yun Liu[2], Pradeep K. Bhaskar[3], Christine Rushlow[2], Erika A. Bach[1]*

**1** Department of Biochemistry and Molecular Pharmacology, New York University Grossman School of Medicine, New York, New York, United States of America, **2** Department of Biology, New York University, New York, New York, United States of America, **3** Laboratory of Biochemistry and Genetics, National Institute of Diabetes and Digestive and Kidney Diseases, National Institutes of Health, Bethesda, Maryland, United States of America

* erika.bach@nyu.edu

## Abstract

Maintenance of somatic sex identity is essential for adult tissue function. In the *Drosophila* testis, adult somatic stem cells known as cyst stem cells (CySCs) require the transcription factor Chinmo to preserve male identity. Loss of Chinmo leads to reprogramming of CySCs into their ovarian counterparts through induction of the female-specific RNA-binding protein Transformer$^F$ (Tra$^F$), though the underlying mechanism has remained unclear. Here, we identify the pioneer transcription factor Zelda (Zld) as a critical mediator of this sex reversal. In wild-type CySCs, *zld* mRNA is repressed by microRNAs (miRs), but following Chinmo loss, these miRs are downregulated, allowing *zld* mRNA to be translated. Zld is necessary for feminization of *chinmo*-mutant CySCs, and ectopic expression of Zld in wild-type CySCs is sufficient to induce Tra$^F$ and drive female reprogramming. Two Zld target genes, *qkr58E-2* and *Ecdysone receptor* (*EcR*), are upregulated in *chinmo*-mutant CySCs and are normally female-biased in adult gonads. Qkr58E-2 facilitates Tra$^F$ production, while EcR promotes female gene expression programs. Zld overexpression feminizes otherwise wild-type CySCs by upregulating EcR, which in turn downregulates the *chinmo* gene. Strikingly, overexpression of Zld also feminizes adult male adipose tissue by inducing Tra$^F$ and downregulating Chinmo, indicating that Zld can override male identity in multiple adult XY tissues. Together, these findings uncover a post-transcriptional mechanism in which miRs-mediated repression of a pioneer factor safeguards male identity and prevents inappropriate activation of the female program in adult somatic cells.

**Data availability statement:** All relevant data are within the paper and its Supporting information files.

**Funding:** Funding to EAB, Grant number: R01-GM085075, Funder: NIH/NIGMS, Funder website: https://www.nigms.nih.gov/; Funding to EAB, Grant number: R35-GM156624, Funder: NIH/NIGMS, Funder website: https://www.nigms.nih.gov/; Funding to CR, Grant number: R35-GM148241, Funder: NIH/NIGMS, Funder website: https://www.nigms.nih.gov/; Funding to CR, Grant number: R01-GM063024, Funder: NIH/NIGMS, Funder website: https://www.nigms.nih.gov/; Funding to CR, Grant number: R03-HD080158, Funder: NIH/NICHD Funder website: https://www.nichd.nih.gov/. The sponsors or funders did not play any role in the study design, data collection and analysis, decision to publish, or preparation of the manuscript.

**Competing interests:** The authors have declared that no competing interests exist.

Abbreviations: BSA, bovine serum albumin; CySCs, cyst stem cells; FCs, follicle cells; FSCs, follicle stem cells; HCR-FISH, hybridization chain reaction fluorescent in situ hybridization; miRs, microRNAs; PBS, phosphate-buffered saline; RBP, RNA-binding protein; RNAi, RNA interference; ROIs, regions of interest; WT, wild-type.

## Introduction

Sex differences are observed in many adult organs during homeostasis and underlie sex-biased responses to injury and disease [1–5]. Phenotypic differences in male and female soma arise during development as a result of several sex-biased processes, including sex chromosome constitution, dosage compensation, and steroid hormones [6–8]. How sex differences are maintained in adult tissues is poorly understood, but gaining such knowledge is important for understanding sex-specific biological responses. The fruit fly *Drosophila* has long been an excellent model for studying sex determination during development and in adulthood [9–11]. Work in flies has shown that sex-specific functions of adult organs such as gut and gonads are maintained by sex-biased splicing cascades, metabolism and steroid hormone responses [1,12–15].

In *Drosophila*, alternative splicing establishes sex-specific somatic differences based on sex chromosome constitution. In flies, XX animals are female, while XY animals are male. XX flies express the RNA-binding protein (RBP) Sex-lethal (Sxl) [16]. Sxl binds directly to exon 2 in *transformer* (*tra*) pre-mRNA, resulting in skipping of exon 2, which contains an early stop codon, and synthesis of full-length Tra (termed Tra$^F$) in females [17,18]. XY flies lack Sxl protein, and as a result, *tra* mRNA incorporates exon 2, resulting in premature translational termination and a presumptive small peptide with no known function [18]. In XX flies, Tra$^F$ alternatively splices *doublesex* (*dsx*) pre-mRNA to produce the female-specific Dsx$^F$ isoform [19–21]. As XY flies lack Tra$^F$, *dsx* pre-mRNA is default-spliced and generates the male-specific Dsx$^M$ isoform. Dsx$^F$ and Dsx$^M$ are members of the DMRT family of transcription factors that regulate sex-specific differences in gene expression and external appearance in *Drosophila* [22,23]. DMRTs have been shown to regulate sex determination across the animal kingdom [24].

In gonads, sex-specific somatic identity is essential for fertility and reproduction. In many mammalian species and in fly species such as *Drosophila melanogaster*, sex-specific somatic cells are required to promote the differentiation of sex-specific gametes and to impart sex identity to germline cells [25,26]. In these species, the production of gametes requires that the sex of the germline and the sex of the soma are the same. If somatic support cells of the gonad lose or reverse their sex identity, gonadal dysgenesis occurs with ensuing infertility [14,15,27,28].

*Drosophila* gonads are an excellent system to study the maintenance of somatic sex identity (S1A and S1B Fig). Both gonads have well-defined niches that support germline stem cells which ultimately give rise to gametes [29]. Both have sex-specific somatic stem cells—cyst stem cells (CySCs) in the testis and follicle stem cells (FSCs) in the ovary—whose daughter cells encapsulate the differentiating germ cells. Somatic sex identity in male adult CySCs depends upon Dsx$^M$ and Chinmo, which contains BTB and Zinc-finger domains [14,15,30,31]. Adult ovarian FSCs, on the other hand, do not express Chinmo or Dsx$^M$ [14,15,32]; instead female somatic sex identity is probably regulated by Tra$^F$, as adult ovaries lacking the Tra$^F$ co-factor Tra-2 degenerate, but detailed analyses were not performed [33]. However, the direct requirement for Tra$^F$ and Dsx$^F$ in maintenance of adult follicle cells (FCs) has not yet been reported.

We and others have found that when CySCs in the testis lose expression of Chinmo, they lose "maleness" and become feminized (S1C Fig) [14,15]. This sex reversal occurs without chromosome abnormalities and causes a catastrophic loss of spermatogenesis with ensuing infertility [15]. We previously reported that the loss of Chinmo in CySCs causes male-to-female sex reversal by upregulating Tra$^F$, which then produces Dsx$^F$ instead of Dsx$^M$ [15]. This Dsx isoform switch causes *chinmo*-mutant CySCs to adopt gene expression and morphology of ovarian FCs [34]. We recently reported that these feminized somatic cells in the testis engage cell behaviors characteristic of ovarian FCs, including female-specific incomplete cytokinesis and collective rotational migration [35]. Importantly, Sxl is not involved in *chinmo*-dependent sex reversal [15].

Despite this progress, we still lack mechanistic insights into how the loss of Chinmo in XY somatic cells leads to the upregulation of Tra$^F$, a factor normally only expressed in XX cells [8]. Changes in gene expression and cellular behavior in *chinmo*-deficient CySCs occur within one day of knockdown, suggesting direct reprogramming of male cells into their female counterparts without an intermediate [34,36]. Studies of direct in vivo reprogramming (i.e., transdifferentiation) have shown that induction of transcription factors, particularly those with pioneering activity, is a key event in this process [37,38].

To gain insights into whether transcription factors are induced during sex reversal, we analyzed our published transcriptomic data of *chinmo*-mutant CySCs. A significant fraction of differentially-upregulated genes in *chinmo*-mutant CySCs contain binding sites for the pioneer transcription factor Zelda (Zld) [34], which binds to site-specific motifs in DNA and increases chromatin accessibility [39–44]. Here, we show that Zld protein is low or absent in WT CySCs, despite expressing robust *zld* transcripts, but Zld protein is strongly upregulated during sex reversal after loss of Chinmo. These results indicate post-transcriptional regulation of *zld* mRNA in XY somatic gonadal cells. In WT CySCs, *zld* mRNA translation is repressed by microRNAs (miRs), which are strongly downregulated upon loss of Chinmo. Overexpressing Zld protein in WT CySCs can induce female-specific splicing of Tra$^F$, placing Zld upstream of a key event in sex reversal and showing that Zld can induce female identity. We identify two direct targets of Zld in the embryo - *qkr58E-2* and *Ecdysone receptor* (*EcR*) [45]—as crucial for male-to-female somatic sex reversal. We find that both targets are female-biased in adult somatic gonadal cells and are upregulated in *chinmo*-mutant and in Zld-overexpressing XY somatic cells. We prove that Qkr58E-2, an RBP, mediates *tra* pre-mRNA alternative splicing downstream of Zld. Furthermore, we show that Qkr58E-2 plays important roles in adult ovarian somatic cells. EcR is known to have multiple critical functions in the somatic ovary [46–50], and EcR target genes are significantly increased in *chinmo*-mutant somatic cells, indicating that EcR is active in these transdifferentiating cells. Surprisingly, we find that prolonged overexpression of Zld in XY cells leads to silencing of the *chinmo* gene, and our results show that Zld-induced EcR downregulates Chinmo protein, consistent with prior work [51]. These results demonstrate that Zld can suppress male sex identity in XY gonadal cells. Finally, we prove that ectopic Zld can feminize adult male adipose tissue by inducing Tra$^F$ and downregulating Chinmo. Thus, Zld can instruct female somatic sex identity and override male sex identity in two independent XY adult tissues through induction of female-biased target genes.

## Results

### Zld protein is upregulated in *chinmo*-depleted somatic cells and is required for male-to-female sex reversal

Loss of Chinmo, either through somatic RNA interference (RNAi) or the *chinmo$^{ST}$* allele that causes loss of Chinmo only in adult somatic cells of the testis, triggers male-to-female transdifferentiation [14,15]. We transcriptionally profiled wild-type (WT) and *chinmo*-depleted somatic cells that had been FACS-purified [34]. Bioinformatic analyses revealed that 304 genes were significantly upregulated in *chinmo*-depleted samples [34]. To determine whether there are common transcription factor signatures among these genes, we used iRegulon [52]. This analysis showed that 30% of upregulated genes had binding sites for Zld. Analysis with DAVID (Database for Annotation, Visualization and Integrated Discovery [53,54]) revealed that the most significantly enriched gene ontology terms among the putative Zld target genes were related to

actin organization, epithelial morphogenesis, cytoskeleton organization, consistent with gain of ovarian somatic gene expression (S2 Fig).

To investigate a potential role for Zld in sex reversal, we performed immunofluorescence with a Zld antibody in testes and other tissues. As expected, Zld was expressed in nuclei of epithelial cells in the larval wing imaginal disc (S3A Fig) [55]. The Zld antibody was specific, as Zld expression was lost in *dpp*-positive cells that overexpressed a *zld-RNAi* transgene (S3B Fig). Zld protein was not expressed in most WT testes (Figs 1A, 1C, and S3C), but occasionally we observed faint Zld expression in a few somatic nuclei (S3D Fig). To further investigate Zld expression in WT testes, we used endogenously tagged alleles *sfGFP-zld* [55] and *mNeonGreen (mNG)-zld* [56]. Both showed robust expression in the larval ventral nerve cord and wing imaginal disc (S3J, S3K, S3N, and S3O Fig). However, neither were expressed in WT testes (S3L and S3P Fig), supporting our results of no or very low expression of Zld in WT testes. Somatic *zld* depletion eliminated the Zld signal in all examined WT testes (S3E and S3G Fig), while somatic Zld overexpression resulted in a 20-fold increase in Zld protein (S3F and S3H Fig). The faint expression of Zld in WT testes does not appear to be functionally important because depletion of Zld from WT male somatic cells did not alter the number of CySCs (S3I Fig).

Zld protein was significantly increased in *chinmo*-deficient somatic cells in $chinmo^{ST/ST}$ or *chinmo*-RNAi by the somatic driver *c587-Gal4* in combination with thermo-sensitive inhibitor *tub-Gal80*$^{ts}$ [57] (termed $c587^{ts}$) (3.5-fold in $chinmo^{ST}$ and 4.5-fold in *chinmo-RNAi*) (Fig 1A–1F). The upregulation of Zld in feminizing testicular somatic cells raised the possibility that Zld is required for male-to-female sex reversal. To test this, we somatically co-depleted *zld* and *chinmo* using the somatic driver *traffic jam (tj)-Gal4* and assessed the number of testes that expressed epithelial Fas3 in somatic cells after 10 days of adulthood. This assay (termed the "Fas3 assay") has been used in past studies to measure feminization [14,15,58]. Whereas 0% of control testes expressing two copies of a neutral *UAS* transgene (*tj>lacZ, lacZ*) had epithelial Fas3 outside of the niche, 100% of testes somatically depleted for *chinmo* (*tj>chinmo-i; lacZ*) displayed Fas3-positive, non-niche somatic cells (Fig 1G, 1H, and 1J). Somatic co-depletion of *chinmo* and *zld* significantly reduced feminization (42% in *tj>chinmo-i; zld-i* compared to 100% for *tj>chinmo-i; lacZ*) (Fig 1G–1J). The same results were observed by somatic depletion of *zld* in $chinmo^{ST/ST}$ testes (64% in $c587>chinmo^{ST/ST}$; *zld-i* compared to 100% for $c587>chinmo^{ST/ST}$) (Fig 1J). While *sfGFP-zld* and *mNG-zld* are homozygous viable, they act as hypomorphs in the adult testis as either allele significantly suppressed feminization in *chinmo*-deficient somatic cells (87% for the former and 80% for the latter, S3M and S3Q Fig); these are the most significant repressions of *chinmo*-dependent feminization that we have ever observed. These results support our conclusion that Zld is required in *chinmo*-deficient cells for feminization.

The upregulation of Zld in *chinmo*-mutant CySCs might reflect the normal female sex determination program in ovarian FCs, but we ruled out this possibility because Zld protein—as monitored by Zld antibody or tagged alleles—was not expressed in the larval or adult ovary or in the larval testis, despite strong expression in the larval ventral nerve cord dissected from the same animals (S4A–S4D Fig). Furthermore, Zld depletion in adult ovarian somatic cells did not impede oogenesis (S4E Fig). Taken together, these results indicate that somatic loss of Chinmo significantly upregulates Zld. While Zld is dispensable for maintenance of adult male somatic cells, Zld is required for conversion of adult male somatic gonadal cells to their female counterparts.

## Multiple miRNAs target *zld* mRNA transcripts in WT male somatic cells

We expected to see upregulation of *zld* mRNA in our RNA-seq analysis of *chinmo*-mutant somatic cells, but *zld* transcripts were unchanged. To validate this result, we used hybridization chain reaction fluorescent in situ hybridization (HCR-FISH) to quantify *zld* transcripts. As predicted, *zld* transcripts were unaltered in *chinmo*-mutant somatic cells compared with WT somatic cells (Fig 2A–2C). These results suggest that *zld* is post-transcriptionally regulated in *chinmo*-mutant feminized cells.

We hypothesized that miRNAs repress *zld* mRNA in WT CySCs. To test this, we depleted the miRNA-processing enzyme Dicer-1 (Dcr-1) in somatic cells and then assessed Zld protein. Somatic depletion of Dcr-1 using two independent

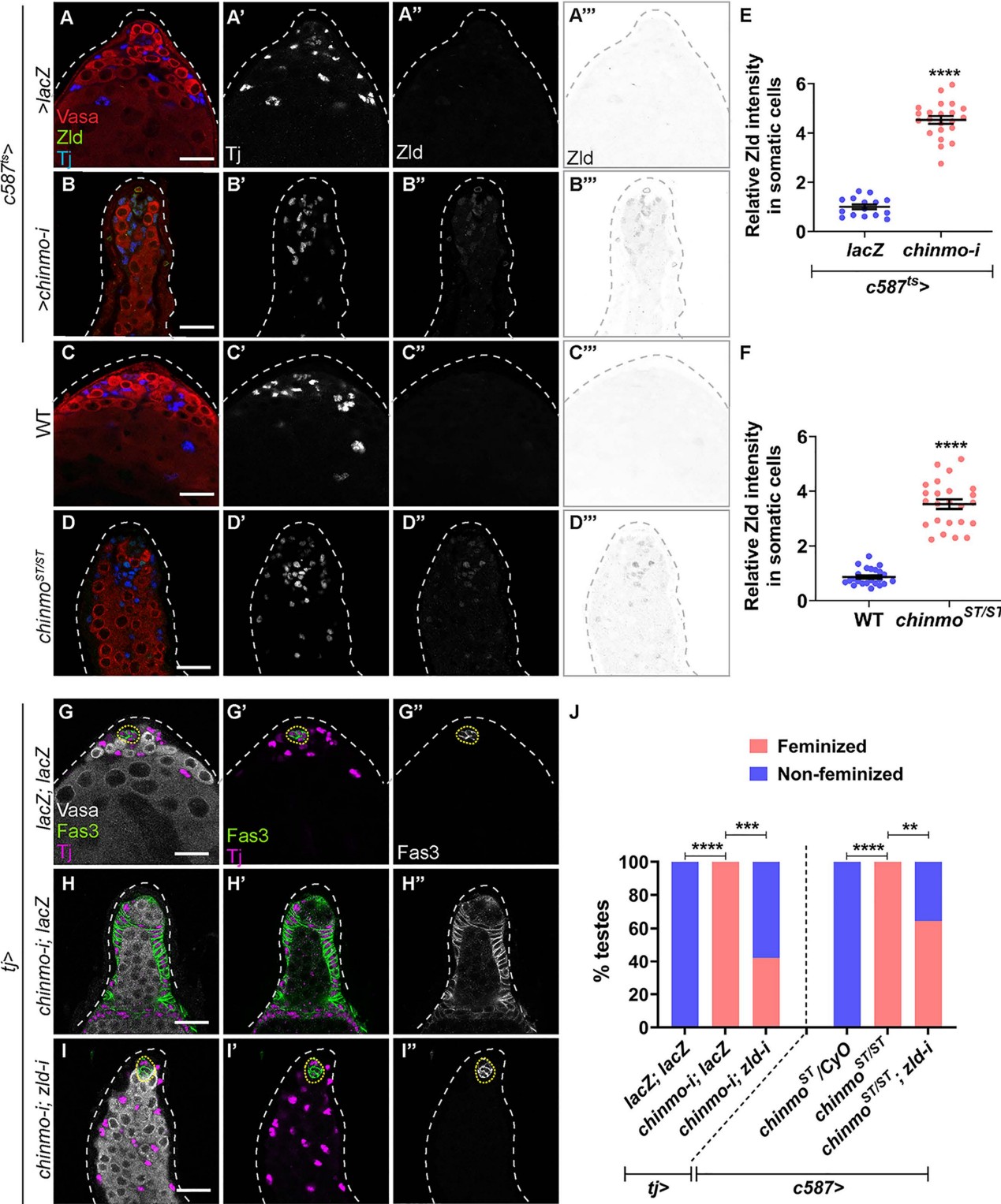

**Fig 1. Zld is upregulated in *chinmo*-depleted somatic cells and is required for feminization. (A–D)** Representative confocal images of control (*c587ts>lacZ*) **(A)**, *c587ts>chinmo-i* **(B)**, WT (Oregon-R) **(C)**, and *chinmoST/ST* **(D)** testes stained for Vasa (red), Tj (blue, grayscale), and Zld (green, grayscale, and inverted grayscale). **(E, F)** Graph showing relative Zld protein expression in somatic cells of *c587ts>lacZ* (*n* = 10) and *c587ts>chinmo-i*

($n=14$), WT ($n=10$), and *chinmo*<sup>ST/ST</sup>

(*n* = 14), WT (*n* = 10), and *chinmo*$^{ST/ST}$ mutant (*n* = 14) testes. **(G–I)** Representative confocal images of a control testis (*tj > lacZ; lacZ*) **(G)**, a testis somatically depleted of *chinmo* (*tj > chinmo-i; lacZ*) **(H)**, and a testis with somatic co-depletion of *chinmo* and *zld* (*tj > chinmo-i; zld-i*) **(I)**. Testes are stained for Vasa (gray), Fas3 (green, grayscale), and Tj (magenta). **(J, left side)** Graph showing the percentage of feminized (pink) and non-feminized (blue) testes in *tj > lacZ; lacZ* (*n* = 10), *tj > chinmo-i; lacZ* (*n* = 13), and *tj > chinmo-i; zld-i* (*n* = 45). **(J, right side)** Graph showing the percentage of feminized and non-feminized testes in *c587 > chinmo*$^{ST}$/*CyO* (*n* = 10), *c587 > chinmo*$^{ST/ST}$ (*n* = 20), and *c587 > chinmo*$^{ST/ST}$; *zld-i* (*n* = 30). Dotted lines mark the niche. Dot plots show individual data points with lines indicating the mean ± SD **(E, F)**. Bar graphs depict the percentage of testes exhibiting the indicated phenotypes **(J)**. The data underlying the graphs shown in the figure can be found in S1 Data. Statistical analysis was performed using Student *t* test (E, F) and Fisher's exact test **(J)** (\*\* *P* = 0.0016; \*\*\* *P* = 0.0002; \*\*\*\* *P* < 0.0001). Scale bars: 20 μm.

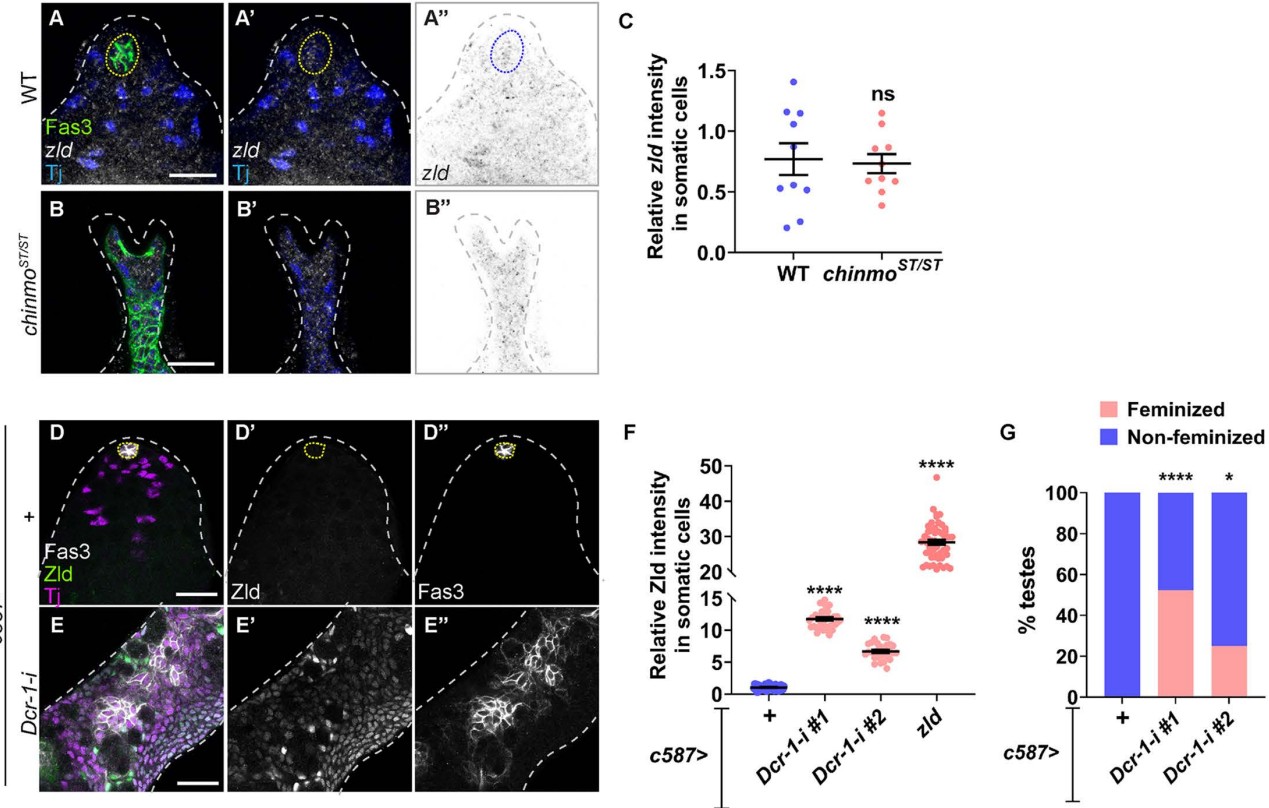

**Fig 2. *zld* is post-transcriptionally regulated in *chinmo*-deficient somatic cells. (A, B)** Representative confocal images of HCR-FISH for *zld* mRNA in WT (Oregon-R) **(A)** and *chinmo*$^{ST/ST}$ testes **(B)**. Testes are stained for Fas3 (green), Tj (blue), and *zld* mRNA (grayscale, inverted grayscale). **(C)** Graph showing no change in the relative *zld* mRNA intensity in somatic cells of WT (*n* = 8) and *chinmo*$^{ST/ST}$ feminized testes (*n* = 10). **(D, E)** Representative confocal images of *c587 > +* **(D)** and *c587 > Dcr-1-i* **(E)** testes stained for Zld (green, grayscale), Fas3 (grayscale), and Tj (magenta). **(F)** Graph showing relative Zld protein expression in somatic cells of *c587 > +* (*n* = 20), *c587 > Dcr-1-i #1* (*n* = 23), *c587 > Dcr-1-i #2* (*n* = 12), and *c587 > zld* (*n* = 15) testes. **(G)** Graph showing the percentage of feminized (pink) and non-feminized (blue) testes in *c587 > +* (*n* = 20), *c587 > Dcr-1-i #1* (*n* = 23), and *c587 > Dcr-1-i #2* (*n* = 12). Dotted lines mark the niche. Dot plots show individual data points with lines indicating the mean ± SD **(C, F)**. Bar graphs depict the percentage of testes exhibiting the indicated phenotypes **(G)**. The data underlying the graphs shown in the figure can be found in S1 Data. Statistical analysis was performed using Student *t* test **(C, F)** and Fisher's exact test **(G)** (ns = not significant; \* *P* = 0.0444; \*\*\*\* *P* < 0.0001). Scale bars: 20 μm.

RNAi transgenes resulted in significant increases in Zld protein (12-fold and 6-fold, respectively, compared to WT somatic cells) (Fig 2D–2F). We are confident in these results because control testes (*c587 > +*) had very low levels of Zld, while overexpression of Zld (*c587 > zld*) caused a 30-fold increase in Zld protein (Fig 2F). Strikingly, somatic depletion of Dcr-1 was sufficient to induce feminization: while no control testes (*c587 > +*) showed Fas3-positive epithelial cells outside the

niche, 52% of *c587 > Dcr-1-i#1*, and 25% of *c587 > Dcr-1-i#2* testes displayed epithelial Fas3-positive somatic cells (Fig 2D, 2E, and 2G). We confirmed the specificity of the Dcr-1 RNAi lines by showing that Dcr-1 protein expression was reduced 0.5-fold in somatic cells overexpressing either transgene (S5A–S5C Fig).

We searched for potential miRNAs targeting the *zld 3'UTR* with TargetScan [59], which predicted seven potential miRNA binding sites: two predicted sites for *miR-263a* (Flybase: *bereft*) and one predicted site each for *miR-316*, *let-7*, *miR-955*, *miR-283*, *miR-125*, and *miR-1011* (Fig 3A).

Based on these observations, we hypothesized that Chinmo (i.e., male sex identity) promotes the expression of one or more miRNAs that in turn repress *zld* mRNA translation in the somatic lineage. To test this model, we performed HCR-FISH using probes with initiators targeting the primary miRNA transcripts. We then assessed which miRNAs are positively regulated by Chinmo using HCR-FISH and which miRNAs repressed Zld protein using immunofluorescence. We did not examine *let-7* since prior work had shown that it is not expressed in WT CySCs [32]. The HCR-FISH analysis revealed that of the remaining six miRNAs, only *miR-1011, miR-263a,* and *miR-283* transcripts were present in WT testes (Fig 3B–3D and 3K–3M) and were significantly reduced in testis somatically depleted for *chinmo* (Fig 3E–3G and 3K–3M). The probes were specific to miRNA transcripts as *miR-1011, miR-263a* and *miR-283* were abolished in testes from *miR-1011KO* or *miR-283KO* homozygous mutant adults and in testes from adults that impaired *miR-263a* somatically through the use of a miR-sponge (SP) (*c587 > miR-263aSP* (Fig 3H–3M)). miR-sponges express miRNA binding sites and are designed to sequester miRNAs and thus knock-down endogenous miRNA activity [60].

To test whether *zld* is a functional target of *miR-1011, miR-263a*, and *miR-283*, we measured *zld* transcript levels in somatic cells of WT and miRNA mutants and sponges using HCR-FISH (Fig 3N–3Q). *zld* transcripts increased 1.5-fold in *miR-1011KO*, 1.72-fold in *miR-263aSP*, and 1.5-fold in *miR-283KO* mutants (Fig 3R), consistent with regulation by these miRNAs.

Taken together, these results suggest that in WT male somatic cells, *zld* mRNA transcripts are silenced by at least three miRNAs whose expression is dependent on male somatic sex identity.

### *miR-1011* and *miR-263a* suppress Zld protein expression to prevent feminization in XY somatic gonadal cells

Since *zld* mRNA was significantly increased in testes from *miR-1011* or *−283* single mutants or from *c587 > miR-263a* sponge, we next tested the prediction that Zld protein would also be increased. Indeed, immunostaining revealed widespread Zld protein upregulation in all three backgrounds (9.6-fold in *miR-1011KO* and 3-fold in *miR-283KO* compared to WT controls, and 9.2-fold in *c587 > miR-263aSP* compared to the scrambled miR-sponge (*c587 > ScrambleSP*)) (Fig 4A–4E, and 4L).

We next wanted to determine whether Zld protein could be further increased by the loss of two or three of these miRs. We originally wanted to test the *miR-1011KO* in combination with *miR-263aSP*; however, we were unable to obtain adult males homozygous for *miR-1011KO* flies carrying *UAS-miR-263a* sponge. Instead, we decided to employ miR-sponge combinations of all three miRNAs. In double miR-sponge combinations, the level of Zld did not increase above that observed with the single *miR-263a* sponge (Fig 4D–4J, and 4L). Similarly, the triple miR-sponge combination (*c587 > miR-283SP; miR-263aSP; miR-1011SP*) also did not surpass the level of Zld in the *miR-263a* sponge background (Fig 4K and 4L). Taken together, these results suggest that *miR-1011* and *miR-263a* exert strong regulatory effects on *zld* mRNA, while *miR-283* contributes weak effects.

We tested whether testes from single miRNA mutants or miR-sponge combinations induced feminization. Epithelial Fas3 outside the niche appeared in 25% of *miR-1011KO* and 20% of *c587 > miR-263aSP* testes (Fig 4B, 4E, and 4M). No feminization was observed in testes from *miR-283KO* or in testes from *miR-283SP* and *miR-1011SP* backgrounds (Fig 4F, 4G, and 4M). Feminization of testes from double and triple miR-sponge background remained at ~20% value observed with the single *miR-263aSP* (Fig 4H, 4I, 4K, and 4M), whereas testes from the *miR-283SP; miR-1011SP* background displayed no feminization (Fig 4J and 4M). Thus, the testes with the highest level of Zld protein could be somatically sex reversed.

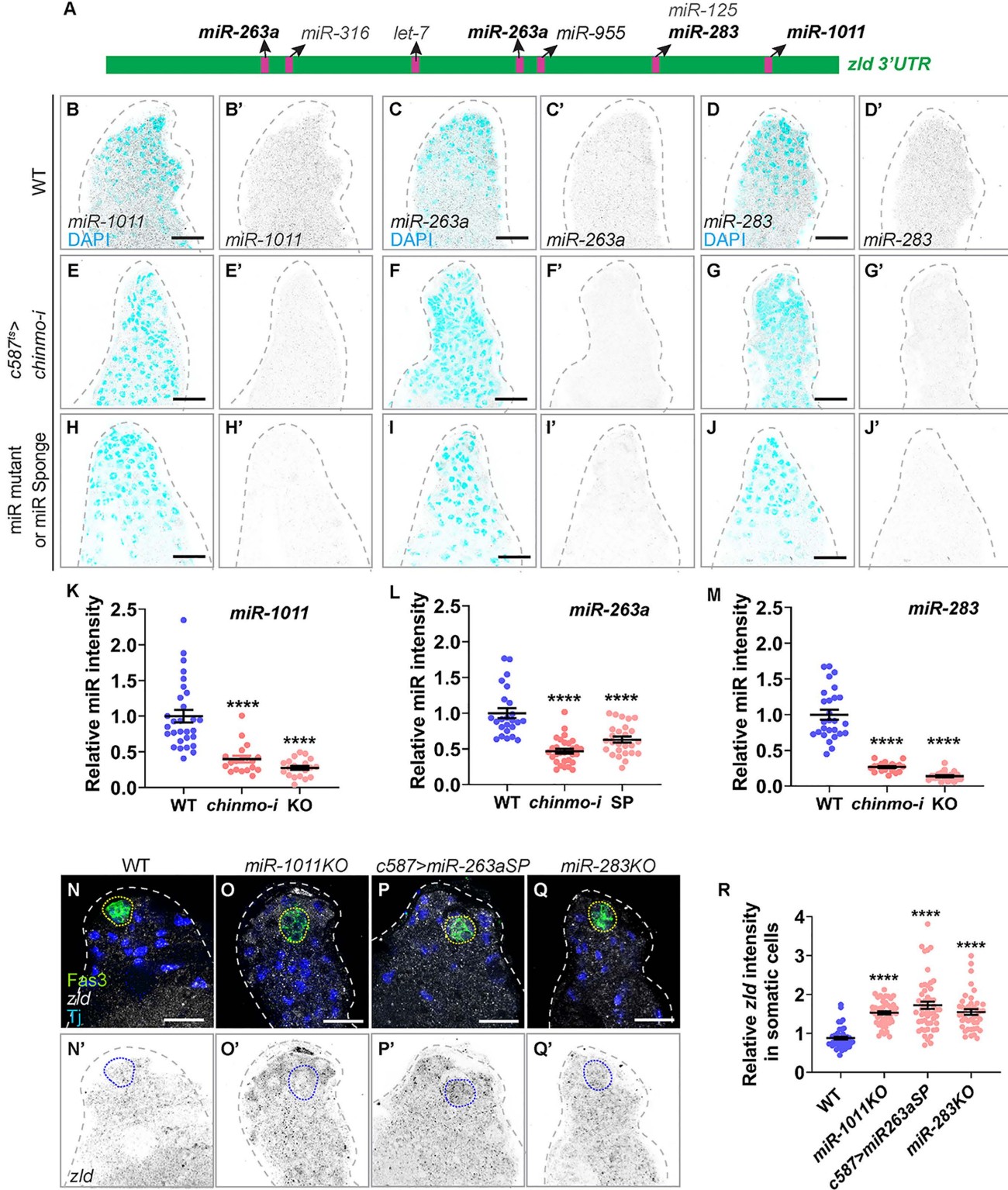

**Fig 3. *miR-1011, miR-263a,* and *miR-283* are expressed in male somatic gonadal cells and repress *zld* mRNA. (A)** Schematic representation of *zld 3'UTR* illustrating the predicted binding sites of candidate miRNAs as identified by TargetScan. **(B–J)** Representative confocal images of HCR-FISH for *miR-1011, miR-263a,* and *miR-283* in WT **(B–D)**, *c587ts>chinmo-i* **(E–G)**, and miR-mutant or Sponge testes **(H–J)**. miRNAs are depicted in

inverted grayscale and DAPI is shown in cyan. **(K–M)** Graph showing relative *miR-1011* intensity in WT ($n = 24$), *c587$^{ts}$>chinmo-i* ($n = 13$), *miR-1011KO* mutant ($n = 10$) testes **(K)**; *miR-263a* intensity in WT ($n = 15$), *c587$^{ts}$>chinmo-i* ($n = 13$), *tj>miR-263aSP* ($n = 15$) **(L)** testes; *miR-283* intensity in WT ($n = 14$), *c587$^{ts}$>chinmo-i* ($n = 8$), *miR-283KO* mutant ($n = 12$) **(M)** testes. **(N–Q)** Representative confocal images of HCR-FISH for *zld* mRNA in WT **(N)**, *miR-1011KO* mutant **(O)**, *c587>miR-263aSP* **(P)**, and *miR-283KO* mutant **(Q)** testes stained for Fas3 (green), Tj (blue), and *zld* mRNA (grayscale and inverted grayscale). **(R)** Graph showing the relative *zld* mRNA intensity in somatic cells of WT ($n = 12$), *miR-1011KO* mutant ($n = 10$), *c587>miR-263aSP* ($n = 12$), and *miR-283KO* mutant ($n = 12$) testes. Dotted lines mark the niche. Dot plots show individual data points with lines indicating the mean ± SD **(K, L, M, R)**. The data underlying the graphs shown in the figure can be found in S1 Data. Statistical analysis was performed using Student *t* test (K, L, M, R) (**** $P < 0.0001$). Scale bars: 20 µm.

Although testes from *miR-283KO* mutants contained significant increases in Zld protein, they did not feminize. However, these testes did have a significant increase in the number of CySCs, defined as Zfh1-positive, Eya-negative cells (S5D–S5F Fig). Furthermore, whereas WT CySCs reside on average 25 µM from the niche, CySCs mutant for *miR-283* reside significantly farther away ($P < 0.001$) (S5G Fig). These data suggest that there is delayed CySC differentiation in *miR-283KO* mutants.

The effects observed in testes from *miR-1011KO* or *miR-283KO* were due to the loss of the miRNA and not to loss of the gene in which it is nested (*Ir93a* and *Gmap*, respectively). Somatic depletion of either *Ir93a* or *Gmap* did not increase Zld expression or alter the number of CySCs (S6A–S6D Fig).

Finally, we assessed whether increasing the level of more than one miRs could impede feminization of *chinmo*-mutant somatic cells. Co-overexpression of *miR-263a* and *miR-1011* significantly reduced the number of *chinmo*-mutant testes undergoing feminization (Fig 5A–5D), while overexpression of these miRs individually did not (Fig 5D). Thus, *miR-1011* and *miR-263a* maintain male identity by suppressing Zld expression, thereby preventing the activation of the female program.

## Zld is sufficient to activate female sex identity in somatic gonadal cells

We investigated whether gain of Zld is sufficient to trigger feminization by monitoring the female sex determinant, Tra$^F$, using a reporter that produces GFP expression when *tra* pre-mRNA undergoes alternative splicing (*UAS-traF$^{\Delta T2AGFP}$*) (Fig 6A) [15]. We previously showed that Tra$^F$ is absent from WT CySCs but is present in WT ovarian FCs and in *chinmo*-mutant CySCs [15]. To assess whether ectopic Zld could induce *tra* pre-mRNA alternative splicing, we overexpressed the Tra$^F$ sensor and *zld* in the adult somatic lineage (*c587$^{ts}$>traF$^{\Delta T2AGFP}$, zld*). Flies were reared at 18°C and upshifted to the restrictive temperature of 29°C after eclosion. Tra$^F$ was observed within 3–4 days of *zld* overexpression in the adult male soma and was fully penetrant, indicating that this critical female-specific event in sex identity was occurring (Fig 6B–6D). We note that ectopic Zld (*c587$^{ts}$>traF$^{\Delta T2AGFP}$, zld*) induces Tra$^F$ more rapidly than loss of Chinmo (*c587$^{ts}$>traF$^{\Delta T2AGFP}$, chinmo-i*), which requires 7–8 days (Fig 6E and 6F), presumably because more Zld was present in the former. Tra$^F$ splicing was significantly increased in Zld overexpressing somatic cells compared to WT male somatic cells but did not reach the level of splicing in WT ovarian FCs (Fig 6G). Overexpression of Zld in XY somatic cells also induced epithelial Fas3 expression outside of the niche, indicating feminization (Fig 6B, 6C, and 6H). At 3–4 days after induction, somatic cells with ectopic Zld still expressed Chinmo (Fig 6I and 6J, arrows), indicating that Zld induces feminization even in the presence of male determinants. Furthermore, acquisition of Tra$^F$ precedes expression of Fas3 in sex-converting somatic gonadal cells as 100% of *c587$^{ts}$>traF$^{\Delta T2AGFP}$, zld* testes exhibited Tra$^F$ expression while only 42% showed epithelial Fas3 expression in non-niche cells (Fig 6D and 6H). These results demonstrate that Zld overrides the male program and activates the female program by inducing first Tra$^F$ and then Fas3.

## Tra$^F$ activation by Zld depends on Qkr58E-2, a conserved KH domain-containing RBP

Since Zld is a transcription factor, it is unlikely to participate directly in alternative splicing of *tra* pre-mRNA. Instead, we reasoned that there must be an RBP that is regulated by Zld. To identify this RBP, we compared 576 annotated RBPs from

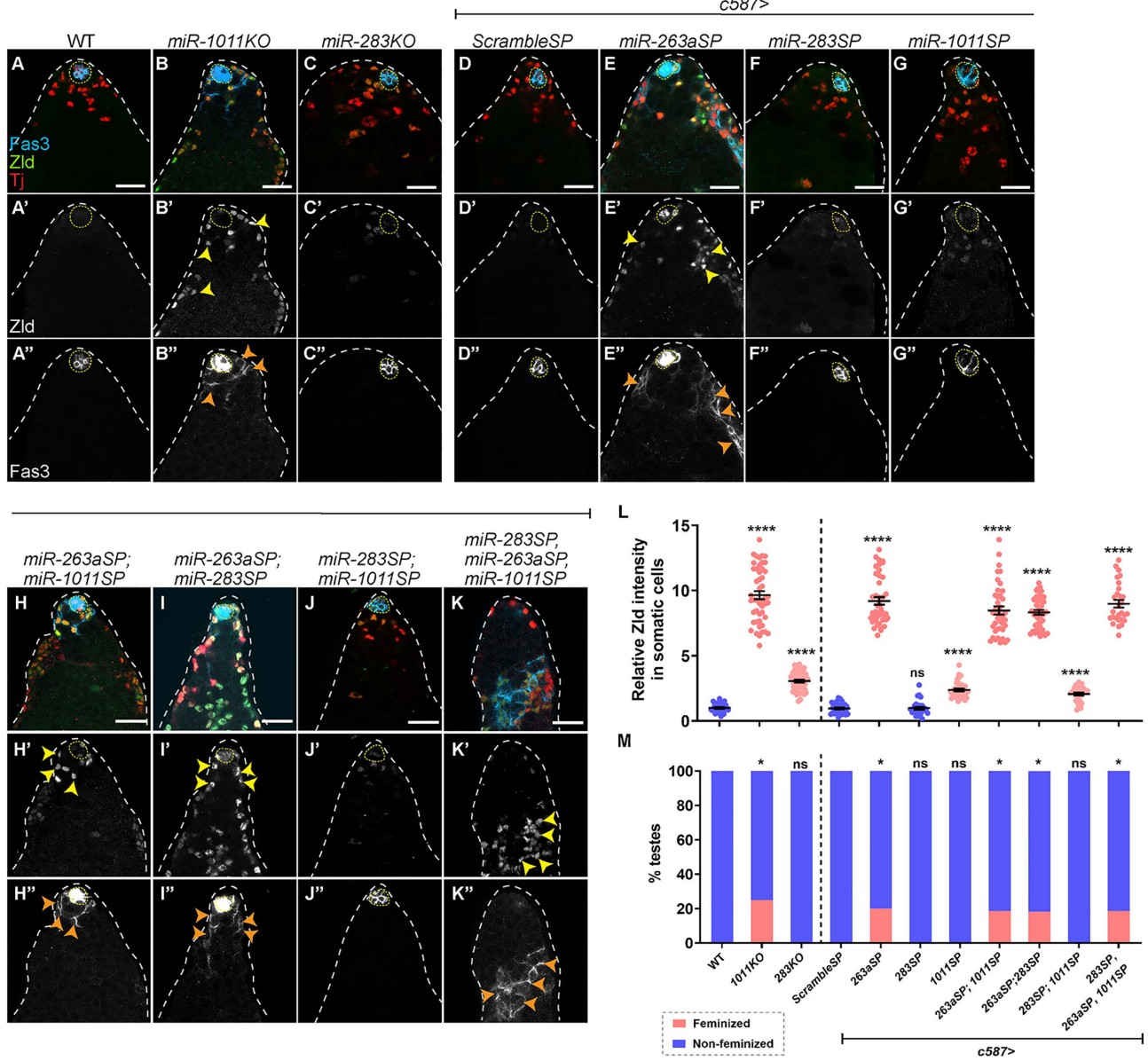

**Fig 4. Zld protein and feminization are induced in male somatic gonadal cells lacking *miR-263a* or *miR-1011*. (A-K)** Representative confocal images of WT **(A)**, *miR-1011KO* **(B)**, *miR-283KO* **(C)**, *c587 > ScrambleSP* **(D)**, *c587 > miR-263aSP* **(E)**, *c587 > miR-283SP* **(F)**, *c587 > miR-1011SP* **(G)**, *c587 > miR-263aSP; miR-1011SP* **(H)**, *c587 > miR-263aSP; miR-283SP* **(I)**, *c587 > miR-283SP; miR-1011SP* **(J)**, and *c587 > miR-283SP; miR-263aSP; miR-1011SP* **(K)** testes at 25 days post induction stained for Tj (red), Zld (green, grayscale) and Fas3 (cyan, grayscale). Yellow arrowheads indicate increased Zld expression and orange arrowheads depict epithelial Fas3-positive cells. **(L, M)** Graph showing relative Zld protein expression **(L)** and percentage of feminized (pink) and non-feminized (blue) testes **(M)** in WT (*n* = 20), *miR-1011KO* (*n* = 12), *miR-283KO* (*n* = 15), *c587 > ScrambleSP* (*n* = 20), *c587 > miR-263aSP* (*n* = 15), *c587 > miR-283SP* (*n* = 11), *c587 > miR-1011SP* (*n* = 12), *c587 > miR-263aSP; miR-1011SP* (*n* = 16), *c587 > miR-263aSP; miR-283SP* (*n* = 11), *c587 > miR-283SP; miR-1011SP* (*n* = 10), and *c587 > miR-283SP; miR-263aSP; miR-1011SP* (*n* = 16) testes. Dotted lines mark the niche. Dot plots show individual data points with lines indicating the mean ± SD **(L)**. Bar graphs depict the percentage of testes exhibiting the indicated phenotypes **(M)**. The data underlying the graphs shown in the figure can be found in S1 Data. Statistical analysis was performed using Student *t* test **(L)** and Fisher's exact test **(M)** (ns = not significant; * *P* = 0.0188 (WT vs. *miR-1011KO*); * *P* = 0.0365 (*c587 > ScrambleSP* vs. *c587 > miR-263aSP*); * *P* = 0.0431 (*c587 > ScrambleSP* vs. *c587 > miR-263aSP; miR-1011SP*); * *P* = 0.0487 (*c587 > ScrambleSP* vs. *c587 > miR-263aSP; miR-283SP*); * *P* = 0.0431 (*c587 > ScrambleSP* vs. *c587 > miR-283SP; miR-263aSP; miR-1011SP*); **** *P* < 0.0001). Scale bars: 20 μm.

PLOS Biology

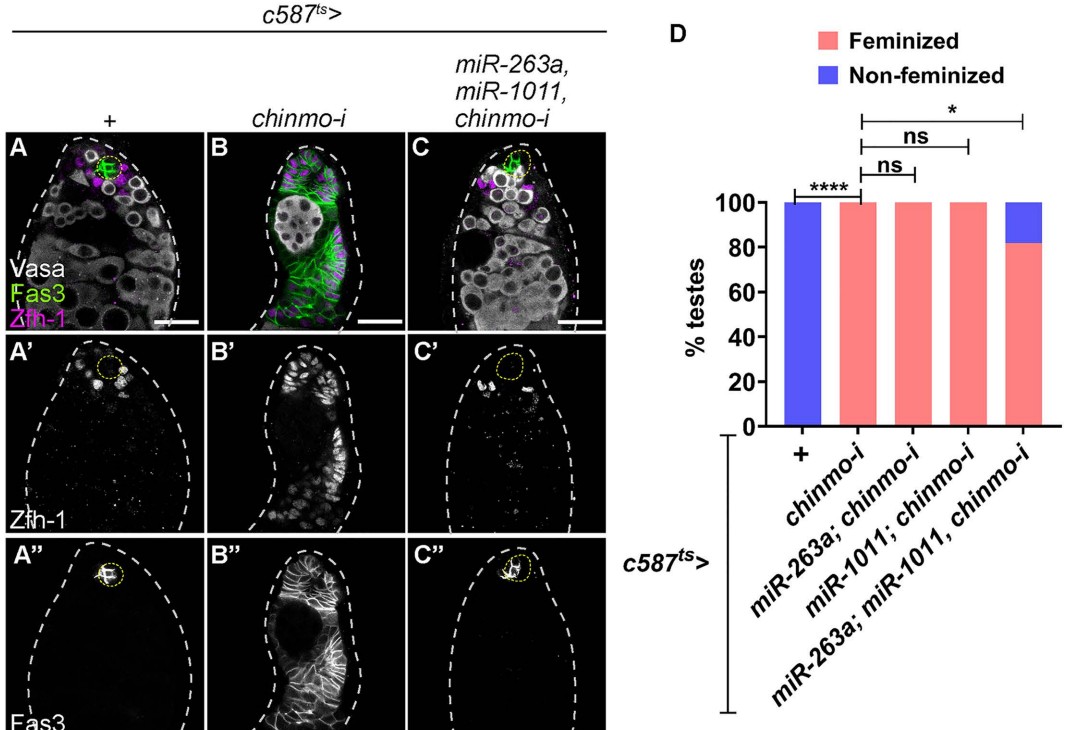

**Fig 5. Overexpression of *miR-263a* and *miR-1011* significantly inhibits feminization of *chinmo*-mutant somatic cells. (A-C)** Representative confocal images of $c587^{ts} >+$ (control) **(A)**, $c587^{ts} > chinmo\text{-}i$ **(B)**, and $c587^{ts} > miR\text{-}263a/miR\text{-}1011; chinmo\text{-}i$ **(C)** testes stained for Vasa (gray), Zfh-1 (magenta, grayscale), and Fas3 (green, grayscale). **(D)** Graph showing the percentage of feminized (pink) and non-feminized (blue) testes in $c587^{ts} >+$ ($n = 10$), $c587^{ts} > chinmo\text{-}i$ ($n = 20$), $c587^{ts} > miR\text{-}263a; chinmo\text{-}i$ ($n = 15$), $c587^{ts} > miR\text{-}1011; chinmo\text{-}i$ ($n = 18$), and $c587^{ts} > miR\text{-}263a/miR\text{-}1011; chinmo\text{-}i$ ($n = 22$). Dotted lines mark the niche. Bar graphs depict the percentage of testes exhibiting the indicated phenotypes **(D)**. The data underlying the graphs shown in the figure can be found in S1 Data. Statistical analysis was performed using Fisher's exact test **(D)** (ns = not significant; * $P = 0.0258$; **** $P < 0.0001$). Scale bars: 20 μm.

the Gene List Annotation for *Drosophila* (GLAD) database [61], to chromatin-immunoprecipitation (ChIP) targets of embryonic Zld [45], and to significantly upregulated genes in *chinmo*-deficient CySCs [34]. This analysis identified Qkr58E-2, which is known to mediate alternative splicing in vitro [62]. Due to a lack of existing Qkr58E-2 antibodies or protein traps, we used HCR-FISH to monitor *qkr58E-2* mRNA in adult gonads. We found that *qkr58E-2* transcripts were significantly increased (1.9-fold) in *chinmo*-mutant somatic cells (Fig 7A–7C), consistent with our RNA-seq results [34]. *qkr58E-2* transcripts displayed female-biased expression in gonads with 2.5-fold higher levels in the somatic ovary than the somatic testis (Fig 7D, 7E, and 7H). *qkr58E-2* transcripts were significantly upregulated (2-fold) in Zld-overexpressing somatic testis at 3–4 days post-induction (Fig 7F–7H), the time point at which Tra^F is robustly expressed (Fig 6C). We proved the specificity of two *qkr58E-2* RNAi lines because each significantly reduced *qkr58E-2* transcripts when expressed in the somatic ovary (S7A–S7C Fig). Importantly, Qkr58E-2 is required for Zld-dependent induction of Tra^F. Tra^F was induced in 100% of testes somatically overexpressing Zld but in only 40% of testes somatically overexpressing Zld and depleting *qkr58E-2* (Fig 7I–7L). Furthermore, *qkr58E-2* is required for feminization as co-depletion of *chinmo* and *qkr58E-2* in adult male somatic cells significantly blocked feminization (Fig 7M).

Qkr58E-2 is a *bona fide* regulator of *tra* pre-mRNA alternative splicing in ovarian FCs, as depleting *qkr58E-2* in these cells reduced both Tra^F and Fas3 expressions in 36% of ovarioles (S8A–S8C Fig). Depletion of *qkr58E-2* in the adult somatic ovary caused severe morphological defects, including fused egg chambers (28.8%), defective germaria (18.7%),

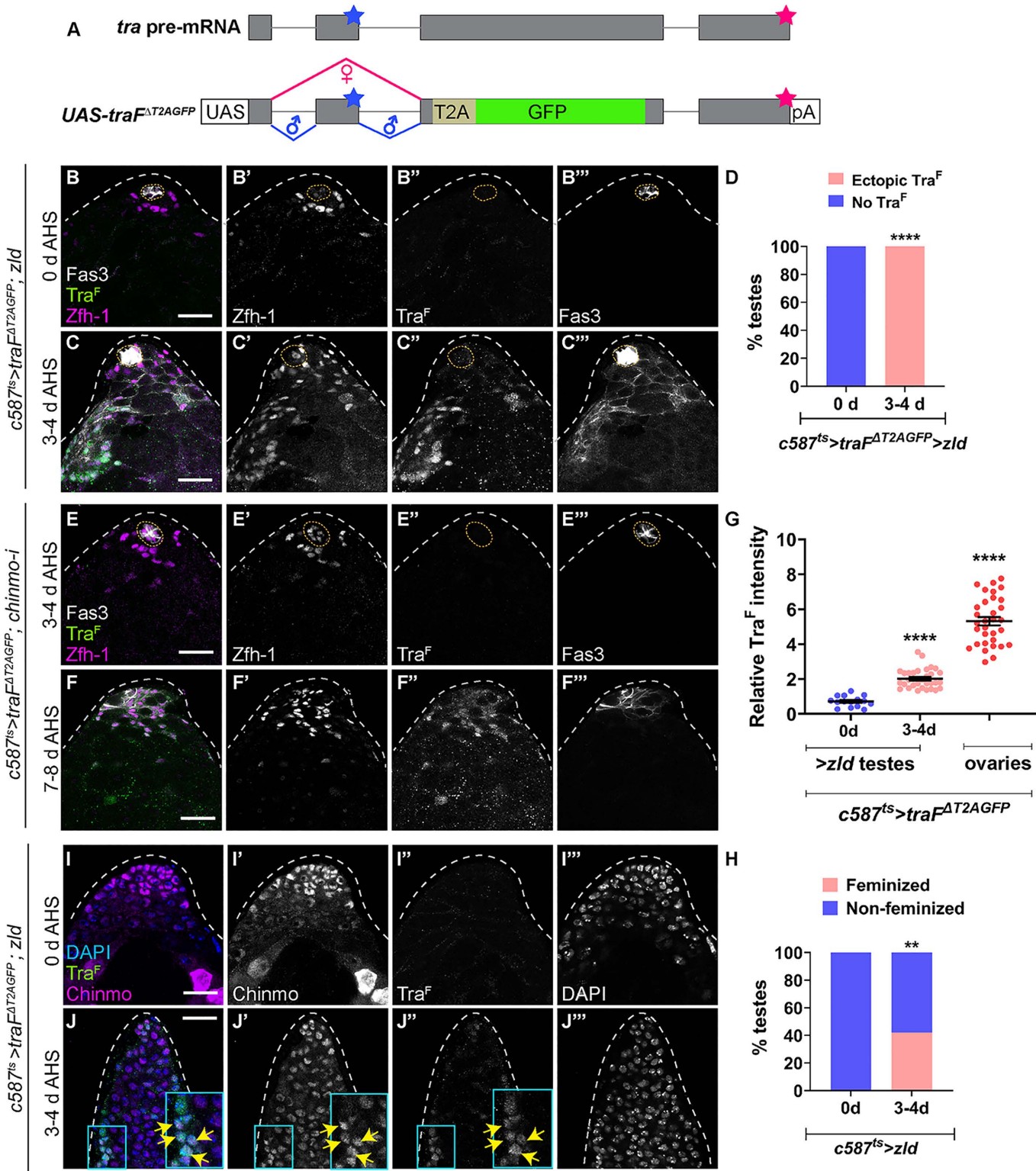

**Fig 6. Ectopic Zld is sufficient to induce feminization. (A)** Schematic of sensor for *tra* pre-mRNA alternative splicing. The *UAS* transgene replaces most of *tra* exon 3 with a T2A-GFP cassette and a polyadenylation signal (pA). Gray boxes denote exons, the blue stars mark stop codon in exon 2, and the red stars mark the stop codon at the end of the sequence. Red lines indicate female-specific splicing, and blue lines show default splicing.

**(B, C)** Representative confocal images of $c587^{ts} > traF^{\Delta T2AGFP}$; *zld* testes at 0 days (uninduced) **(A)** and 3-4 days of adulthood **(B)**. Alternative splicing of *tra* pre-mRNA (labeled Tra$^F$) was monitored using anti-GFP (green, grayscale), Zfh-1 (magenta, grayscale), and Fas3 (grayscale). **(D)** Graph showing the percentage of testes exhibiting ectopic Tra$^F$ (pink) and no Tra$^F$ (blue) in $c587^{ts} > traF^{\Delta T2AGFP}$; *zld* at 0 days ($n = 10$) and 3–4 days ($n = 14$) of adulthood. **(E, F)** Representative confocal images of $c587^{ts} > traF^{\Delta T2AGFP}$; *chinmo-i* testes at 3–4 days **(E)** and 7–8 days **(F)** of adulthood, stained for GFP (indicative of Tra$^F$, green, grayscale), Zfh-1 (magenta, grayscale), and Fas3 (grayscale). **(G)** Graph of relative expression of Tra$^F$ in somatic cells of $c587^{ts} > traF^{\Delta T2AGFP}$; *zld* testes at 0 days ($n = 10$) and 3–4 days ($n = 14$) of adulthood, and $c587^{ts} > traF^{\Delta T2AGFP}$ ovaries ($n = 8$). **(H)** Graph of percentage of feminized (pink) and non-feminized (blue) testes in $c587^{ts} > zld$ at 0 days ($n = 15$) and 3–4 days ($n = 28$) of adulthood. **(I, J)** Representative confocal images of $c587^{ts} > traF^{\Delta T2AGFP}$; *zld* testes at 0 days (uninduced) **(I)** and 3–4 days of adulthood **(J)**, stained for GFP (green, grayscale), DAPI (blue, grayscale), and Chinmo (magenta, grayscale). Arrows in J–J″ in inset show somatic cells overexpressing *zld* that induce Tra$^F$ and express Chinmo. Dotted lines mark the niche. Dot plots show individual data points with lines indicating the mean ± SD **(G)**. Bar graphs depict the percentage of testes exhibiting the indicated phenotypes **(D, H)**. The data underlying the graphs shown in the figure can be found in S1 Data. Statistical analysis was performed using Student *t* test **(G)** and Fisher's exact test **(D, H)** (** $P = 0.0031$; **** $P < 0.0001$). Scale bars: 20 μm.

and increased cell death (52.2%), rendering females infertile (S8D, S8E, and S8G Fig). By contrast, depletion of *qkr58E-2* in adult somatic testes had no discernable effect (S9A–S9C Fig). Somatic depletion of Tra$^F$ target *dsx* in adult ovaries mimicked *qkr58E-2* loss (S8F and S8G Fig), highlighting the necessity of maintaining female sex identity in adulthood for proper ovary function and fertility.

### *qkr58E-2* is sufficient for feminization of XY somatic gonadal cells

To investigate whether *qkr58E-2* is sufficient to induce Tra$^F$ expression, we overexpressed *qkr58E-2* using *qkr58E-2$^{G3095}$*, a P-element containing *UAS* sites inserted in the *5'UTR* of *qkr58E-2*. Indeed, *qkr58E-2* transcripts increased 6-fold in $c587 > qkr58E-2^{G3095}$ (S7D–S7F Fig). Somatic *qkr58E-2$^{G3095}$* overexpression caused a significant 2.9-fold increase in Tra$^F$ compared to WT male somatic cells (Fig 8A–8D). Furthermore, somatic Qkr58E-2 overexpression induced feminization in 45% of testes (Fig 8A–8C and 8E). These results suggest that Qkr58E-2 is female-biased and is a key mediator of *tra* pre-mRNA alternative splicing in *chinmo*-mutant and Zld-overexpressing somatic gonadal cells.

### Sustained ectopic Zld suppresses male identity by silencing the *chinmo* gene

We wanted to assess the outcome of overexpressing Zld for longer than 3–4 days of adulthood (See Fig 6). When Zld was somatically overexpressed for 14 days of adulthood, somatic cells lost Chinmo and gained Castor, a female-specific protein in gonads and a marker of feminization (Fig 9A and 9B) [14,15]. HCR-FISH analysis revealed that 14 days of Zld caused the transcriptional downregulation of the *chinmo* gene as no *chinmo* transcripts were observed in feminized (i.e., Fas3-positive) Zld-overexpressing cells (Fig 9D, cells inside the dashed yellow lines), while robust *chinmo* transcripts were observed in neighboring Zld-overexpressing somatic cells that had not yet feminized (Fig 9D, cells outside the dashed yellow lines). *chinmo* transcripts were unaltered in testes prior to Zld induction (Fig 9C). These results suggest that the feminization caused by ectopic Zld occurs in two successive steps: first ectopic Zld induces the female determinant Tra$^F$ that activates the female program, and then Zld suppresses the male program by downregulating the *chinmo* gene (Fig 9E).

### Zld-induced Chinmo downregulation and feminization requires EcR

We investigated how Zld silences Chinmo. We first tested whether upregulation of female determinants was sufficient to repress Chinmo. However, prolonged overexpression of Tra$^F$ or Dsx$^F$ in the somatic testes for 20 days did not affect Chinmo expression or induce feminization (S10A–S10D Fig), consistent with previous reports [14,15].

We next considered EcR as a candidate because it is a direct Zld target in the embryo [45], *EcR* mRNA is upregulated in *chinmo*-mutant CySCs [34], and EcR causes downregulation of Chinmo in other *Drosophila* tissues [63]. Additionally, the van Doren lab reported that EcR protein exhibits female-biased expression in larval gonads [51]. Using the Ag10.2

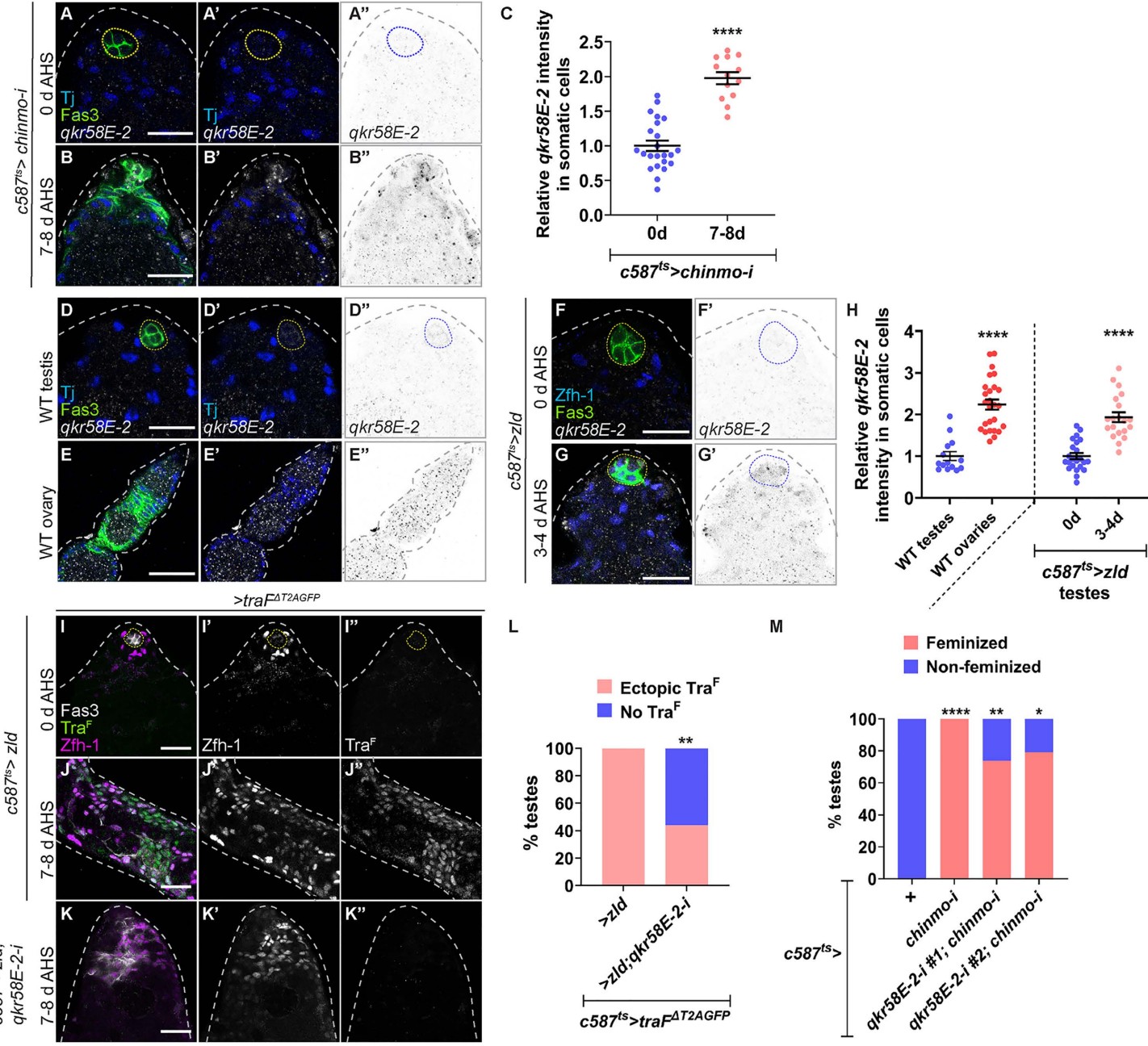

**Fig 7. Qkr58E-2 mediates alternative splicing of *tra* downstream of Zld. (A, B)** Representative confocal images of HCR-FISH for *qkr58E-2* mRNA in *c587ts>chinmo-i* testes at 0 days **(A)** and 7–8 days of adulthood **(B)**, stained for Fas3 (green), Tj (blue), and *qkr58E-2* mRNA (grayscale, inverted grayscale). **(C)** Graph showing relative *qkr58E-2* mRNA intensity in somatic cells of *c587ts>chinmo-i* testes at 0 days (n = 8) and 7–8 days (n = 11) of adulthood. **(D–G)** Representative confocal images of HCR-FISH for *qkr58E-2* mRNA in WT testis **(D)**, WT ovary **(E)**, *c587ts>zld* testes at 0 days (uninduced) **(F)** and 3–4 days of adulthood **(G)**. The gonads are stained for Fas3 (green), Tj (blue in **D, E**), Zfh-1 (blue in **F, G**), and *qkr58E-2* mRNA (grayscale, inverted grayscale). **(H)** Graph showing relative *qkr58E-2* mRNA intensity in somatic cells in WT testes (n = 8), WT ovaries (n = 8), and *c587ts>zld* testes at 0 days (n = 8) and 3–4 days (n = 12) of adulthood. **(I–K)** Representative confocal images of *c587ts>traF^{ΔT2AGFP}; zld* testes at 0 days (uninduced) **(I)** and at 7–8 days **(J)**, and *c587ts>traF^{ΔT2AGF}; zld; qkr58E-2-i* at 7–8 days of adulthood **(K)** stained for GFP (indicative of Tra^F) (green, grayscale) and Zfh-1 (magenta, grayscale). **(L)** Graph showing the percentage of testes with ectopic Tra^F (pink) and no Tra^F (blue) in *c587ts>traF^{ΔT2AGFP}; zld* (n = 14) and *c587ts>traF^{ΔT2AGFP}; zld; qkr58E-2-i* (n = 11). **(M)** Graph showing the percentage of feminized (pink) and non-feminized (blue) testes in *c587ts>+* (n = 20), *c587ts>chinmo-i* (n = 25), *c587ts>qkr58E-2-i #1; chinmo-i* (n = 19), and *c587ts>qkr58E-2-i #2; chinmo-i* (n = 19). Dotted lines mark the niche. Dot plots show individual data points with lines indicating the mean ± SD **(C, H)**. Bar graphs depict the percentage of testes exhibiting the indicated phenotypes **(L, M)**. The data underlying the graphs shown in the figure can be found in S1 Data. Statistical analysis was performed using Student *t* test **(C, H)** and Fisher's exact test **(L, M)**. (* *P* = 0.0286; ** *P* = 0.0026 **(L)**; ** *P* = 0.0107 **(M)**; **** *P* < 0.0001). Scale bars: 20 μm.

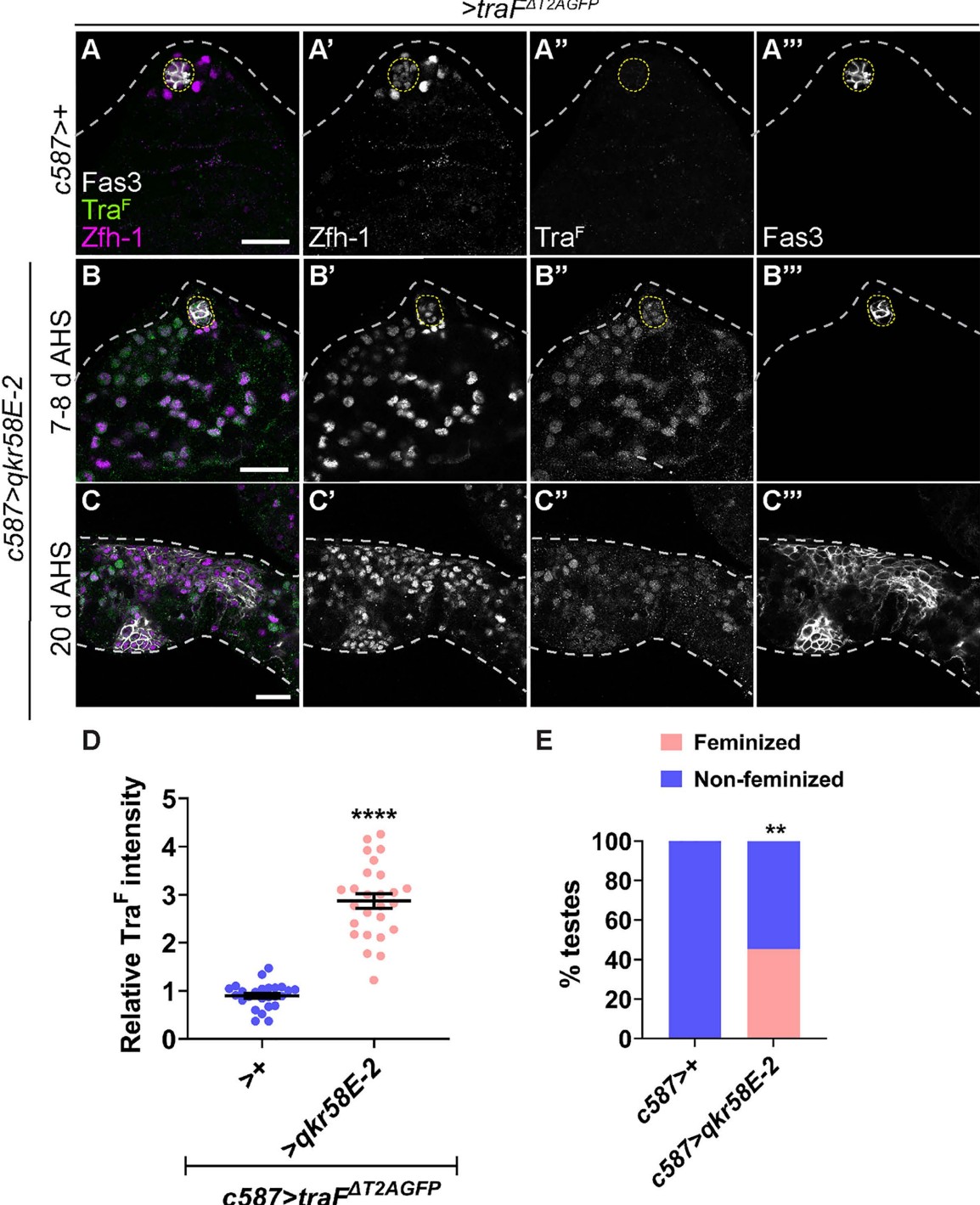

**Fig 8. Overexpression of *qkr58E-2* is sufficient to feminize male somatic gonadal cells. (A–C)** Representative confocal images of *c587 > traF^ΔT2AGFP^ >+* **(A)** and *c587 > traF^ΔT2AGFP^*; *qkr58E-2* testes at 7−8 days **(B)** and 20 days **(C)** of adulthood stained for GFP (indicative of Tra^F^) (green, grayscale), Zfh-1 (magenta, grayscale), and Fas3 (grayscale). **(D)** Graph showing relative Tra^F^ expression in somatic cells of *c587 > traF^ΔT2AGFP^ >+* (*n* = 9) and *c587 > traF^ΔT2AGFP^*; *qkr58E-2* (*n* = 10) testes. **(E)** Graph showing the percentage of feminized (pink) and non-feminized (blue) testes in *c587 >+* (*n* = 15) and *c587 > qkr58E-2* (*n* = 11). Dotted lines mark the niche. Dot plots show individual data points with lines indicating the mean ± SD **(D)**. Bar graphs depict the percentage of testes exhibiting the indicated phenotypes **(E)**. The data underlying the graphs shown in the figure can be found in S1 Data. Statistical analysis was performed using Student *t* test **(D)** and Fisher's exact test **(E)** (** *P* = 0.007; **** *P* < 0.0001). Scale bars: 20 μm.

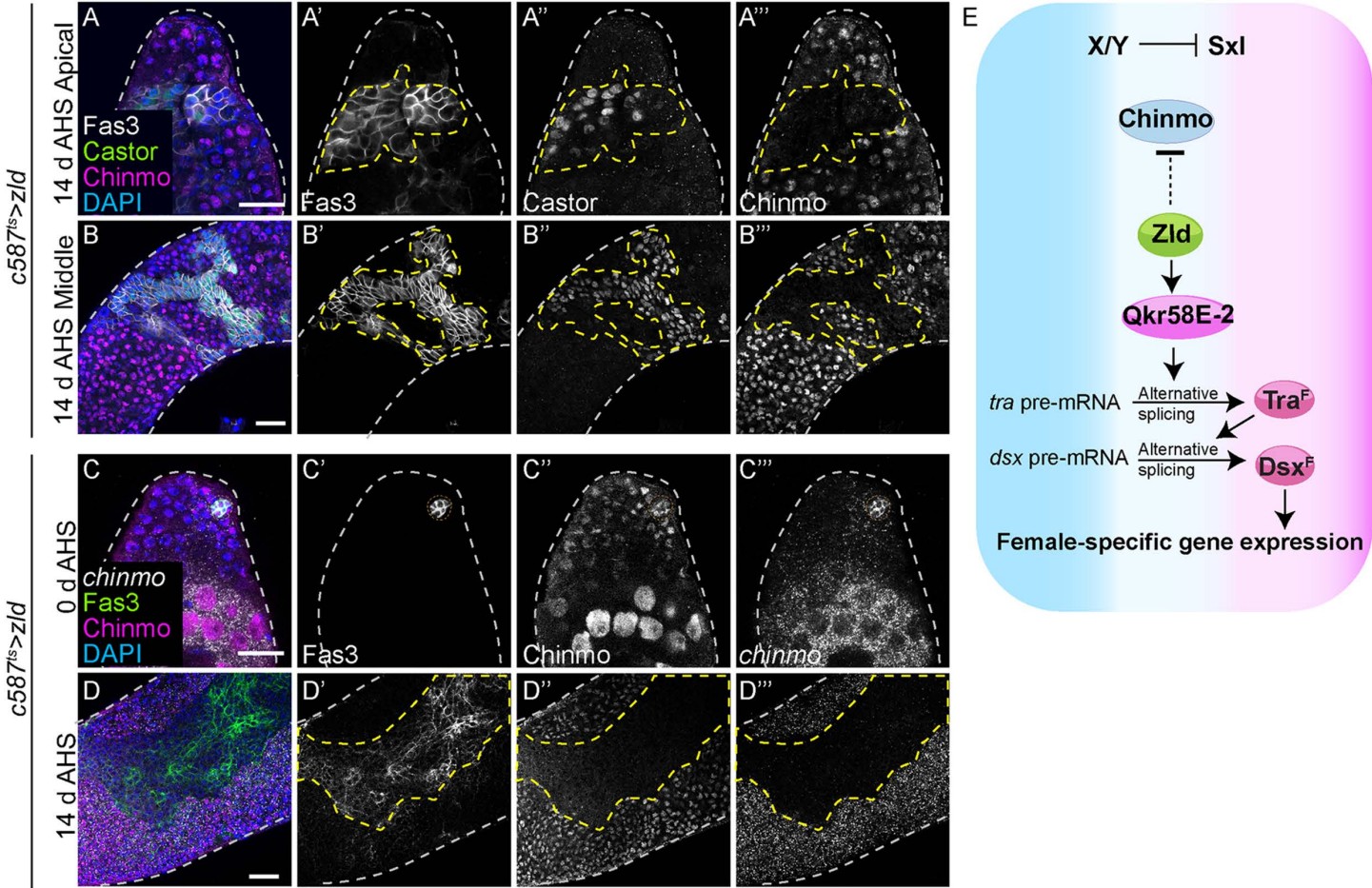

**Fig 9. Sustained overexpression of Zld silences the *chinmo* gene. (A, B)** Representative confocal images of the apical **(A)** and middle parts **(B)** of *c587^{ts} > zld* testes at 14 days of adulthood stained for Fas3 (grayscale), Castor (green, grayscale), Chinmo (magenta, grayscale), and DAPI (blue). Yellow dotted lines mark feminization. **(C, D)** Representative confocal images of HCR-FISH for *chinmo* mRNA in *c587^{ts} > zld* testes at 0 days (uninduced) **(C)** and 14 days of adulthood **(D)**, stained for Fas3 (green, grayscale), DAPI (blue), Chinmo (magenta, grayscale), and *chinmo* mRNA (grayscale). Yellow dotted lines mark feminization. **(E)** Model: Ectopic Zld regulates two steps in feminization of a male somatic cell: (1) Zld triggers alternative splicing of *tra*-pre mRNA to produce Tra^F, which activates female program; and (2) Zld suppresses Chinmo expression, which inhibits the male program. Dotted lines mark the niche. Scale bars: 20 μm.

antibody that recognizes all EcR isoforms, we found that EcR was present in WT adult ovarian FCs. This expression was lost when EcR was depleted in adult ovarian FCs (*c587^{ts} > EcR-i*), confirming antibody specificity (Fig 10A, 10B, and 10E). Using the same antibody master mix and the same confocal settings, we found EcR was absent from CySCs and early cyst cells but was occasionally observed in late cyst cells in WT testes (Fig 10C, arrows). Indeed, EcR levels were 3-fold higher in the somatic ovary than testis (Fig 10E). Consistent with upregulation of *EcR* mRNA in *chinmo*-mutant CySCs, EcR protein was significantly upregulated in these cells (Fig 10D and 10E). Furthermore, known EcR target genes were significantly increased in *chinmo*-mutant CySCs, including *Hr3* (30-fold $P^{adj} < 0.034$) and *Eip63F-1* (5.4-fold $P^{adj} < 0.095$), indicating that EcR signaling occurs in *chinmo*-mutant CySCs.

We tested whether Zld overexpression induces EcR in male somatic cells. Because *chinmo* loss also induces EcR (Fig 10D and 10E), we analyzed EcR expression at a time point when Chinmo was still present (i.e., 10 days of Zld overexpression). Ectopic Zld significantly increased EcR in most somatic cells at this time point (Fig 10F–10H). As expected,

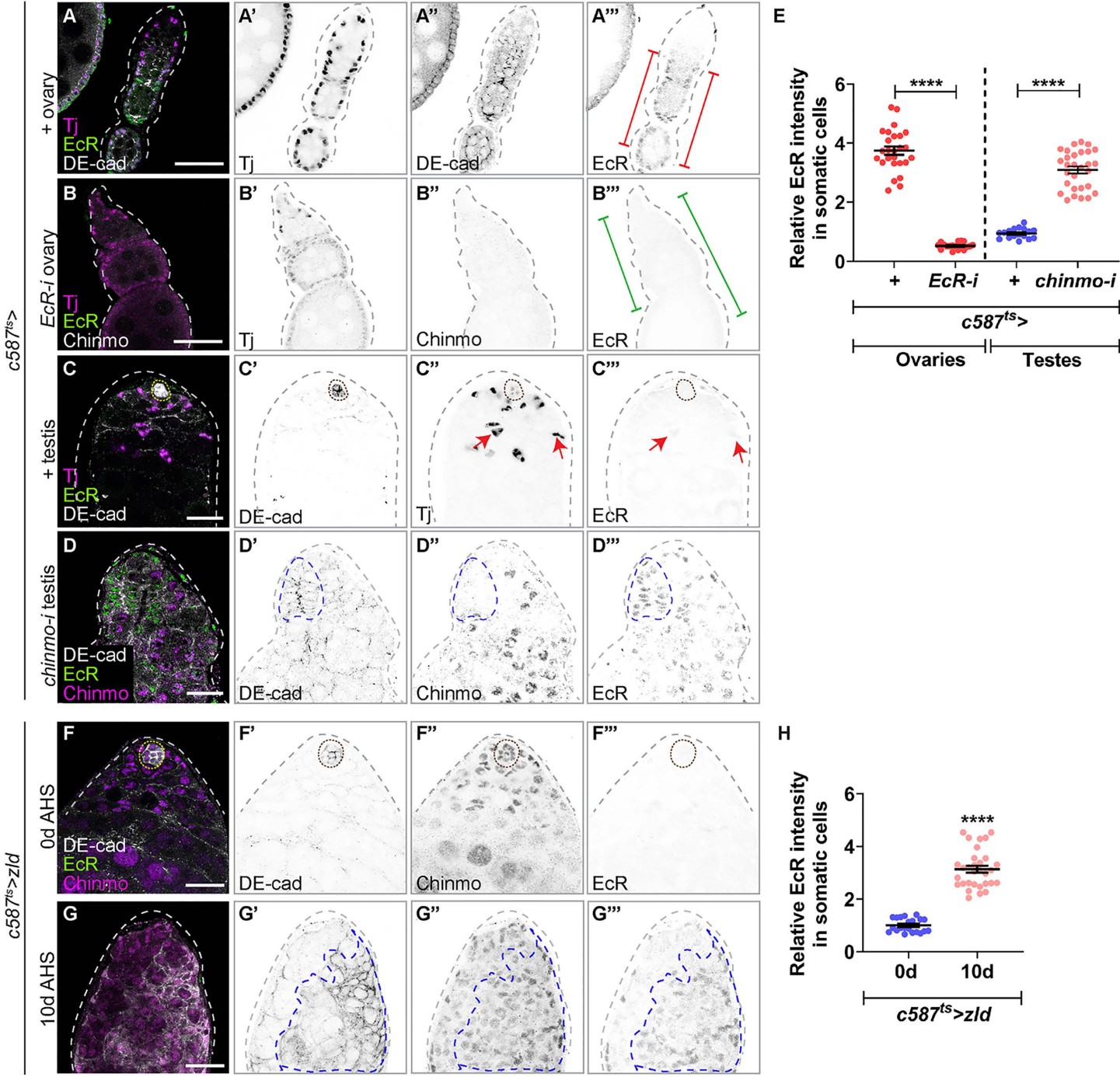

**Fig 10. Zld overexpression in male somatic gonadal cells upregulates EcR. (A–D)** Representative confocal images of $c587^{ts} > +$ ovary **(A)**, $c587^{ts} > EcR\text{-}i$ ovary **(B)**, $c587^{ts} > +$ testis **(C)**, and $c587^{ts} > chinmo\text{-}i$ testis stained for DE-cadherin (DE-Cad, gray, inverted grayscale in **A**, **C**, **D**), EcR (green, inverted grayscale in **A–D**), Tj (magenta, inverted grayscale in **A–C**) and Chinmo (magenta, inverted grayscale in **B**, **D**). Red brackets in **A″′** show robust EcR protein in ovarian follicle cells. Green brackets in **B″′** show diminished expression of EcR in the follicle cells. Red arrows in **C″** and **C″′** show faint EcR expression in late differentiated cyst cells. Blue dashed line in **D′–D″′** show ectopic EcR expression in *chinmo*-depleted somatic cells. **(E)** Graph showing relative EcR intensity in somatic cells in $c587^{ts} > +$ ovaries ($n = 8$), $c587^{ts} > EcR\text{-}i$ ovaries ($n = 12$), $c587^{ts} > +$ testes ($n = 8$), and $c587^{ts} > chinmo\text{-}i$ testes ($n = 8$). **(F, G)** Representative confocal images of $c587^{ts} > zld$ testes at 0 days (uninduced) **(F)** and 10 days of adulthood **(G)** stained for DE-cadherin (grayscale, inverted grayscale), Chinmo (magenta, inverted grayscale), and EcR (green, inverted grayscale). Blue dotted lines

in G′-G‴ mark feminized cells. **(H)** Graph showing relative EcR intensity in the somatic cells in *c587^ts^>zld* testes at 0 days (*n* = 8) and 10 days (*n* = 10) of adulthood. Dotted lines mark the niche. Dot plots show individual data points with lines indicating the mean ± SD **(E, H)**. The data underlying the graphs shown in the figure can be found in S1 Data. Statistical analysis was performed using Student *t* test **(E, H)** (**** *P* < 0.0001). Scale bars: 20 μm.

these EcR-positive cells also expressed Chinmo (Fig 10G″). These results indicate that upregulation of EcR, similar to that of Qkr58E-2 and Tra^F^, is an early event in male-to-female sex reversal caused by ectopic Zld.

Finally, we assessed whether Zld-induced Chinmo loss and feminization depend on EcR. Depleting EcR in Zld-overexpressing somatic cells (*c587^ts^>zld; EcR-i*) significantly inhibited both Chinmo loss and feminization: 100% of testes overexpressing Zld showed Chinmo loss and feminization, whereas only 33% did so when EcR was depleted (Fig 11A–11C, 11F, and 11G). Co-overexpression of EcR and its cofactor Taiman (Tai) (*tj>EcR, tai*) was sufficient to trigger Chinmo loss and feminization (Fig 11D–11G), consistent with prior results [51]. Thus, Zld-induced repression of the male program and activation of the female program require EcR (Fig 11H).

### Ectopic Zld feminizes adult male fat body

We next tested whether ectopic Zld could cause sex reversal in other adult male tissues. We chose the adult fat body (i.e., *Drosophila* adipose tissue), which comprises sexually dimorphic polyploid cells. We found that male adipose cells expressed Chinmo and Dsx^M^ (Dsx^M^::GFP) but lacked Tra^F^ and the Dsx^F^ target Yolk protein 1 (Yp1, Yp1::GFP). Female adipose cells expressed Tra^F^ and Yp1, but lacked Chinmo and Dsx^M^ [22,64,65] (Figs 12A–12F, and S11). Male adult fat cells overexpressing Zld induced alternative-splicing of *tra* pre-mRNA (Fig 12G–12I) and significantly downregulated Chinmo protein (Fig 12G and 12J). Thus, ectopic Zld can induce the female sex identity program and repress male identity in two somatic tissues, gonad and fat body.

### Discussion

The key findings of this study (Fig 13) are that (1) Zld is repressed by miRs in WT male CySCs but is translated upon loss of Chinmo; (2) gain of Zld in *chinmo*-mutant CySCs leads to upregulation of Qkr58E-2 and EcR, both of which are female-biased in the somatic gonad; (3) Qkr58E-2 produces Tra^F^, which in turn generates Dsx^F^; (4) EcR promotes female-biased gene expression by upregulating targets like *Hr3* and *Eip63F-1*. Thus, Zld lies upstream of the production of two female-biased transcription factors—Dsx^F^ and EcR—that serve critical roles in the ovary. We further showed that ectopic expression of Zld in the WT male somatic gonad causes feminization through Qkr58E-2 and EcR. Since depletion of *EcR* in Zld-overexpressing cells significantly blocks Chinmo loss and feminization, we favor the interpretation that the induction of EcR in Zld overexpressing CySCs leads to the downregulation of the *chinmo* gene. Finally, we show that ectopic Zld converts adult male adipose tissue to its female counterpart.

Our work demonstrates a critical role for Qkr58E-2 in maintaining female sex identity in adult ovarian FCs. Somatic loss of Qkr58E-2 from adult ovaries leads to defective oogenesis. Interestingly, we show that adult somatic depletion of Dsx^F^ from ovaries yields similar phenotypes; our results showing the requirement for Dsx^F^ in the adult somatic ovary is the first demonstration, to our knowledge, of this. Collectively, these results indicate that female somatic sex identity needs to be maintained in the adult ovary, similar to the need for sustaining male somatic sex identity in the testis [14,15]. Intriguingly, KHDRBS1, the mouse homolog of Qkr58E-2, is required somatically for female fertility in mice. Since human patients with missense mutations in *KHDRBS1* have premature ovarian failure [66–69], it is likely that our work has direct relevance to human fertility.

This study demonstrates that EcR is female-biased in adult somatic gonadal cells. EcR is strongly expressed in ovarian FCs. We observe EcR occasionally in differentiating somatic cells in the testis but not in CySCs. This contrasts with prior work, which reported EcR expression throughout the testicular somatic lineage [70,71]. We are at a loss to reconcile these discrepancies, but we are confident with our results because EcR depletion from FCs significantly reduced EcR

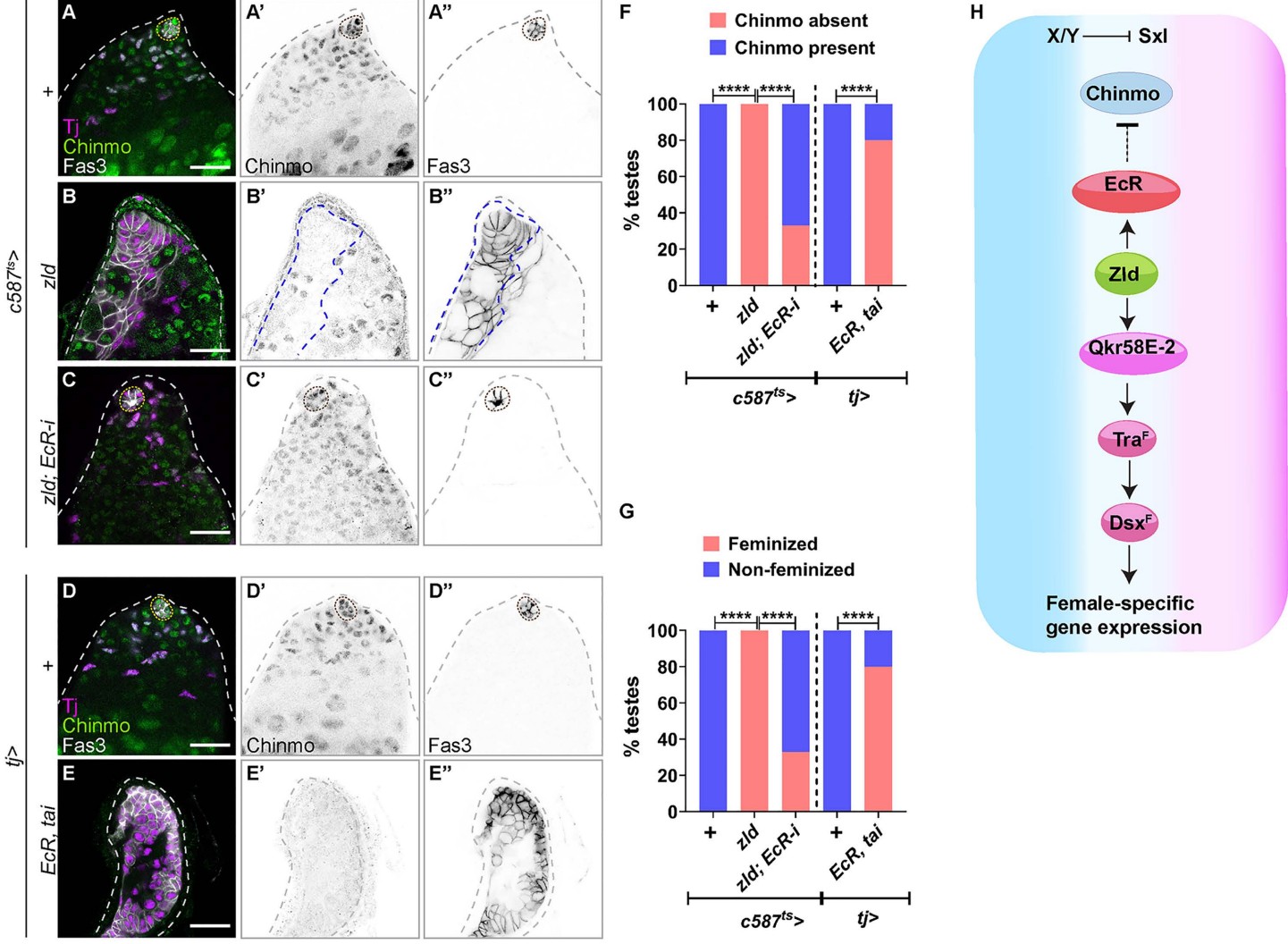

**Fig 11. EcR is required for Chinmo suppression and feminization in Zld-overexpressing somatic cells. (A–C)** Representative confocal images of *c587ts>+* **(A)**, *c587ts>zld* **(B)**, and *c587ts>zld; EcR-i* **(C)** testes at 14 days of adulthood stained for Tj (magenta), Chinmo (green, inverted grayscale), and Fas3 (grayscale, inverted grayscale). **(D, E)** Representative confocal images of *tj>+* **(D)** and *tj>EcR, tai* testes **(E)** stained for Tj (magenta), Chinmo (green, inverted grayscale), and Fas3 (grayscale, inverted grayscale). **(F)** Graph showing the percentage of testes with presence (blue) and absence (pink) of Chinmo in *c587ts>+* (*n* = 15), *c587ts>zld* (*n* = 15), *c587ts>zld; EcR-i* (*n* = 25), *tj>+* (*n* = 15), and *tj>EcR, tai* (*n* = 12). **(G)** Graph showing the percentage of feminized (pink) and non-feminized (blue) testes in *c587ts>+* (*n* = 15), *c587ts>zld* (*n* = 15), *c587ts>zld; EcR-i* (*n* = 25), *tj>+* (*n* = 15), and *tj>EcR, tai* (*n* = 12). **(H)** Model: Zld upregulates EcR, which then suppresses Chinmo. Dotted lines mark the niche. Bar graphs depict the percentage of testes exhibiting the indicated phenotypes **(F, G)**. The data underlying the graphs shown in the figure can be found in S1 Data. Statistical analysis was performed using Fisher's exact test **(F, G)** (**** *P* < 0.0001). Scale bars: 20 μm.

protein (Fig 10A, 10B, and 10E). We used the same EcR antibody master mix to stain both ovaries and testes and the same confocal settings to image EcR expression. Our data suggest that EcR signaling is absent or very low in WT soma of the testis, which is consistent with prior reports showing that EcR is dispensable in WT testes [46,71]. Our model is also consistent with recent work demonstrating that mating induces EcR signaling in CySCs, which then disrupts somatic encystment of male germ cells [72]. Thus, suppressing EcR signaling in the adult male somatic gonad is important for spermatogenesis.

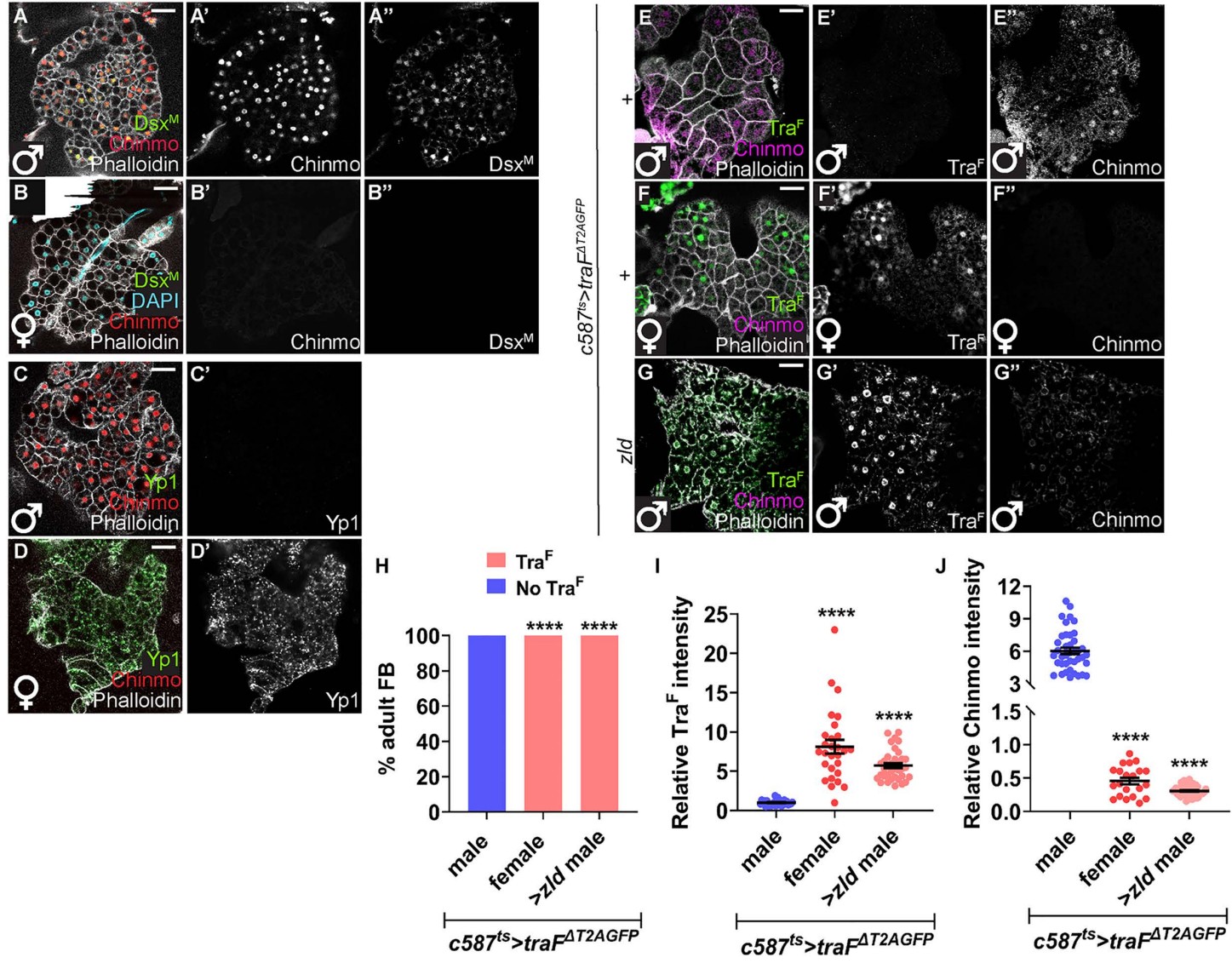

**Fig 12. Ectopic Zld feminizes male adipose tissue. (A, B)** Representative confocal images of male **(A)** and female **(B)** Dsx^M::GFP adult fat body stained for Chinmo (red, grayscale), Phalloidin (grayscale), and Dsx^M (GFP, green, grayscale). **(C, D)** Representative confocal images of male **(C)** and female **(D)** Yp1::GFP adult fat body stained for Chinmo (red), Phalloidin (grayscale), and Yp1 (GFP, green, grayscale). **(E–G)** Representative confocal images of $c587^{ts}>traF^{\Delta T2AGFP}$+ male **(E)**, $c587^{ts}>traF^{\Delta T2AGFP}$+ female **(F)**, and $c587^{ts}>traF^{\Delta T2AGFP}$; *zld* male fat body **(G)** stained for GFP (indicative of Tra^F) (green, grayscale), Phalloidin (grayscale), and Chinmo (magenta, grayscale). **(H)** Graph showing the percentage of testes exhibiting ectopic Tra^F (pink) and no Tra^F (blue) in $c587^{ts}>traF^{\Delta T2AGFP}$ male ($n = 8$), $c587^{ts}>traF^{\Delta T2AGFP}$ female ($n = 7$), and $c587^{ts}>traF^{\Delta T2AGFP}$; *zld* male fat body ($n = 12$). **(I)** Graph showing relative Tra^F expression in $c587^{ts}>traF^{\Delta T2AGFP}$ male ($n = 8$), $c587^{ts}>traF^{\Delta T2AGFP}$ female ($n = 7$), and $c587^{ts}>traF^{\Delta T2AGFP}$; *zld* male fat body ($n = 12$). **(J)** Graph showing relative Chinmo expression in $c587^{ts}>traF^{\Delta T2AGFP}$ male ($n = 8$), $c587^{ts}>traF^{\Delta T2AGFP}$ female ($n = 7$), and $c587^{ts}>traF^{\Delta T2AGFP}$; *zld* male fat body ($n = 12$). Dot plots show individual data points with lines indicating the mean ± SD **(I, J)**. Bar graphs depict the percentage of testes exhibiting the indicated phenotypes **(H)**. The data underlying the graphs shown in the figure can be found in S1 Data. Statistical analysis was performed using Student *t* test **(I, J)** and Fisher's exact test **(H)** (**** $P < 0.0001$). Scale bars: 20 μm.

Our results raise the possibility that EcR directly silences the *chinmo* gene. If true, we would expect to observe EcR binding directly to the *chinmo* locus. This has not yet been reported in Zld-overexpressing male somatic cells. In the larval wing disc, downregulation of *chinmo* during normal development correlated with an increase in EcR, however EcR did not bind the *chinmo* locus, suggesting an indirect effect [73]. EcR may silence the *chinmo* gene in male somatic gonadal cells.

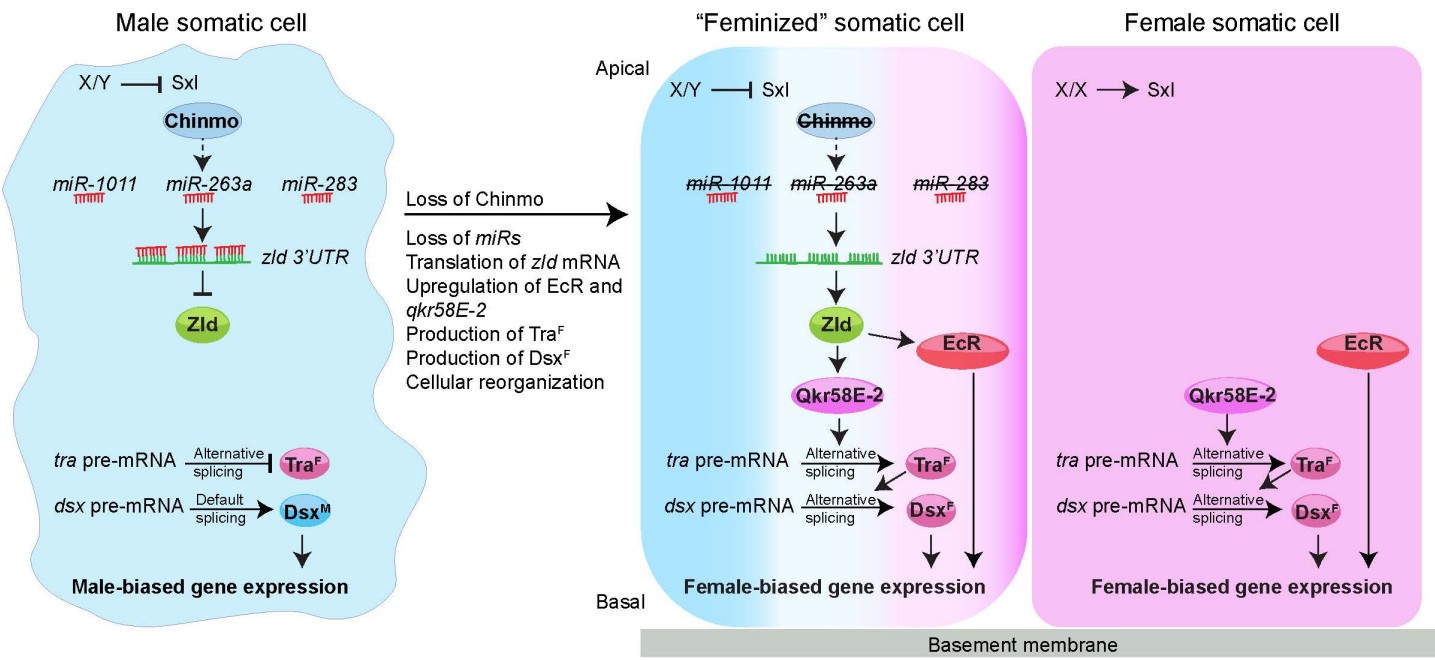

**Fig 13. Model.** (Left, Male somatic cell) In male somatic gonadal cells, Chinmo positively regulates expression of *miR-1011, miR-263a,* and *miR-283*, which suppress *zld* mRNA, thereby maintaining low Zld protein levels and enforcing the male program. (Middle, "Feminized" somatic cell) Loss of Chinmo disrupts this regulatory network, leading to reduced miRNA expression and increased translation of *zld* mRNA. Elevated Zld protein initiates two key feminization processes: (1) activation of Qkr58E-2, which facilitates alternative splicing of female-specific *tra* pre-mRNA to Tra^F, triggering the female program through production of Dsx^F, and (2) upregulation of EcR, a crucial regulator of follicle cell development and oogenesis. (Right, Female somatic cell) In female somatic gonadal cells, Qkr58E-2 promotes alternative splicing of *tra* pre-mRNA to Tra^F. EcR promotes expression of female-biased genes that regulate follicle cell differentiation.

In support of this model, the EcR target Broad (Br) and Chinmo are reciprocal antagonists during development, possibly at the level of transcription [74,75]; however, adult CySCs already express Br [71,72], so this mechanism is unlikely. Another possibility is that EcR promotes expression of *let-7* miRNA, which then downregulates *chinmo-RH* isoform that has a ~6 kb extension of the *3′ UTR*; this type of regulation of *chinmo* has been reported in mushroom body lineage [76]. However, HCR-FISH revealed that the *chinmo-RH* isoform is not expressed in WT somatic cells of the testis [77], making this model unlikely. Finally, it is possible that EcR represses *chinmo* in male somatic gonadal cells through an unknown mechanism; in the larval neuroepithelium, the mechanism by which EcR downregulates Chinmo is also unknown [63]. Future work using CUT&RUN and ATAC-seq will be needed to determine how EcR represses the *chinmo* gene in somatic cells of the testis.

The fact that EcR signaling can repress Chinmo in the adult somatic testis raises the question of whether the lack of Chinmo in the adult somatic ovary is due to the presence of EcR in FCs. Our results suggest that this is not the case since Chinmo is still absent in adult FCs depleted of EcR (Fig 10B″). However, it is possible that EcR represses Chinmo during ovary development, and this repression is maintained in adulthood.

Our work shows that Zld protein is not expressed in the third instar larval or adult ovary. This result was unexpected because Zld promotes female sex identity in the early embryo by promoting expression of the *Sxl* "establishment" promoter that is required for the first burst of Sxl, which then auto-regulates its own expression through the Sxl "maintenance" promoter [16]. This suggests that Zld is required very early in development to induce female somatic sex but is not needed later for sex identity maintenance. Interestingly, the Nystul lab reported that *zld* mRNA was among the top 50 transcripts in FSCs in the adult ovary [78]. Taken together with our data, this suggests that *zld* mRNA may be latent in adult somatic gonadal cells, perhaps as a result of their shared development.

Our study is to our knowledge the first to demonstrate in vivo repression of a pioneer factor by miRs. Prior work from cultured ESCs in vitro has shown that a double-negative feedback loop balances pluripotency and differentiation: miRs repress pioneer factor mRNA and the pioneer factor protein represses transcription of the miRs [79,80]. Here, we show that male identity in somatic cells promotes expression of miRs that are required for normal differentiation of male somatic gonadal cells. When male identity is lost, these miRs are downregulated and the pioneer factor mRNA is translated, which induces female sexual fate. Furthermore, our Dcr-1 and miR-KO results unequivocally demonstrate that somatic miR-NAs—particularly *miR-1011* and *miR-263a*—restrict Zld expression and inhibit activation of the female program. Future work will be needed to determine whether *zld* mRNA is repressed in other adult somatic cells and whether this occurs through *miR-1011*, *-283*, and *-263a*.

We have shown that ectopic Zld can feminize two *dsx*-positive somatic tissues (male gonad and male fat body). However, recent work has shown that Tra$^F$ is functional in all female cells, regardless of *dsx* status [8]. Therefore, it will be important to determine whether ectopic Zld can feminize *dsx*-negative cells, which comprise most cells in the female fly.

While we conclude that Zld is required for sex reversal, it is possible that Zld plays a generic role in cell fate reprogramming and not in sex reversal per se. For example, Zld permits cell fate reprogramming in diverse contexts, including wing disc regeneration where *zld* is dispensable for normal development of the wing disc but essential for its regeneration [81], and in the nervous system where Zld reverts partially differentiated neuroblasts to a stem cell-like state [82]. These findings, together with our results, raise the possibility that Zld acts more broadly by opening chromatin and reactivating silenced genes or by randomly activating a set of its target genes. In our case, feminization occurs from the activation of specific Zld targets *qkr58E-2* and *EcR*. Future studies using chromatin accessibility assays such as ATAC-seq will be essential to delineate the precise repertoire of Zld target genes in our model system.

## Materials and methods

### Experimental animals

*Drosophila melanogaster* strains and RRIDs used in this study are listed in S1 Table. For a list of full genotypes by figure, see S2 Table.

*Drosophila* were reared on food made with these ingredients: 1,800 mL molasses (LabScientific, Catalog no. FLY-8008-16), 266 g agar (Mooragar, Catalog no. 41004), 1,800 g cornmeal (LabScientific, Catalog no. FLY-8010-20), 744g Yeast (LabScientific, Catalog no. FLY-8040-20F), 47 L water, 56 g Tegosept (Sigma no. H3647-1KG), 560 mL reagent alcohol (Fisher no. A962P4), and 190 mL propionic acid (Fisher no. A258500).

Flies were raised at 25°C except crosses with *Gal80$^{ts}$*, which were maintained at 18°C until eclosion, and the adult flies were transferred to 29°C. Lineage-wide overexpression or depletion was achieved using the Gal4/UAS system [83]. The *Gal80$^{ts}$* transgene was used with *c587-Gal4* or *tj-Gal4* to somatically deplete *chinmo*, *Dcr-1*, *qkr58E-2*, *zld*, *EcR*, or *dsx* or to somatically overexpress *zld*, *qkr58E-2*, *tra$^F$*, *dsx$^F$*, *miR-1011SP, miR-263aSP, miR-283SP, scrambledSP, lacZ, miR-1011*, *miR-263a*, or *EcR+tai* in adult male somatic cells. *c587-Gal4* driver is active in the somatic cells of germarium including FSCs that give rise to the adult follicle epithelium [84]. Flies containing multiple transgenes (*UAS-EcR, UAS-tai*) were generated by meiotic recombination using eye color for selection.

### *UAS-zld shmIR (zygotic) 27F* transgene

The *UAS-zld shmIR (zygotic) 27F* (abbreviated *zld*-RNAi) for depleting *zld* zygotically was constructed by inserting the passenger strand sequence 5′-*CAGCAGCTACATCAACAGCTA*-3′ from [41] into the *Valium20* vector [85] and was injected into *Drosophila* embryos.

### *UAS-zld* transgene

A cDNA encoding full-length Zld was subcloned into *UAST* vector [83], which was injected into *Drosophila* embryos by Best Gene

**DsxM::FLAG-EGFP allele (abbreviated DsxᴹGFP)**

We designed a *dsx* locus modification to add FLAG C-terminal peptide tag (GenBank: KX714724.1) to the endogenous Dsxᴹ protein (see S11 Fig). As a reporter of *dsx* expression, we also appended the self-cleaving T2A sequence (GenBank: MW331579.1) to the locus followed by EGFP fluorescent protein coding sequence (GenBank: MN517551.1) and a nuclear localization sequence. Because the EGFP protein is small, it can diffuse back into the cytoplasm [86]. Because the Dsx proteins have a common 5′ encoded DNA-binding domain, followed by 3′ sex-specific splicing events encoding distinct effector domains, the tagged endogenous Dsxᴹ protein should be sex-specific.

The *DSX-M-FLAG-EGFP* DNA cassette was synthesized and cloned into the *pUC57-Brick* vector. We transformed flies using homology-mediated CRISPR by injecting homology DNAs into embryos of a strain expressing the Cas9 endonuclease (BDSC-56552) (*w¹¹¹⁸; PBac[y[+mDint2]=vas-Cas9] VK00037/CyO, P[w[+mC]=Tb[1]]Cpr[CyO-A]*). gRNAs were cloned into *puc19-3xP3-dsred-attb-gypsy-U63-gRNA* and were injected into embryos. We did preliminary screening of possible new *dsx* alleles by crossing single emerging G0 flies to balancer stock (*w+; +/+; TM3,Sb/TM6B,Tb*). G1 Male flies over the balancer chromosome were crossed back to balancer stock (*w+; +/+; TM3,Sb/TM6B,Tb*). Cas9 is marked with 3xP3-GFP which is expressed in the eye, and stocks were selected not to have GFP in eye as an evidence of Cas9 removal. Multiple flies were subjected to PCR screen to identify positive insertion.

We used these primers to amplify PCR product from single fly gDNA. These primers are on either side of the tag.

For left arm: 641bp

DSXM6Fw-cgcataacttctgttaatccccagctcg

DSXMSeqRV1-accaccccggtgaacagctcctcg

Sequenced by: DSXMSeqFw1-aaatcgcactgtagcccagatctac

For right arm:378bp

EGFP-C- CATGGTCCTGCTGGAGTTCGTG

DSXM5Rv- atgtcgatctgttcctcgatttcaa

Sequenced by: DSXMSeqRV2-taaggaacgtaaggaagtgagaac

DSX-M:

cgcataacttctgttaatccccagctcgagtggaaataaatcgcactgtagcccagatctactacaactactacaccccgatggccctggtgaacggggcgcccatgtacctgacctacccgagcatcgaacagggtcgctatggggcgcacttcacccatctgccgctcacacagatttgtccaccgactccagagccgctggccctcagccgctccccgagcagtcccagtggaccgtcggctgtccacaaccaaaagccctcccgaccgggcagcagcaatggcaccgtccactccgcgAcctcacccacaatggtcaccacgatggccacgacctcctccacgcccacgctcagccgccgtcagagatcgcgctcggccacgcccaccactccgccaccaccgccaccggcgcacagcagcagcaacggagcctaccaccacggccaccacctAgtcagctccacggctgccacggactacaaagacgatgacgataaagactacaaagacgatgacgataaagactacaaagacgatgacgataaaGAGGGCAGGGGAAGTCTTCTAACATGCGGGGACGTGGAGGAAAATCCCGGCCCCgtgagcaaggcgaggagctgttcaccggggtggtgcccatcctggtcgagctggacggcgacgtaaacggccacaagttcagcgtgtccggcgagggcgagggcgatgccacctacggcaagctgaccctgaagttcatctgcaccaccggcaagctgcccgtgccctggcccaccctcgtgaccaccctgacctacggcgtgcagtgcttcagccgctacccgaccacatgaagcagcacgacttcttcaagtccgccatgcccgaaggctacgtccaggagcgcaccatcttcttcaaggacgacggcaactacaagacccgcgccgaggtgaagttcgagggcgacaccctggtgaaccgcatcgagctgaagggcatcgacttcaaggaggacggcaacatcctggggcacaagctggagtacaactacaacagccacaacgtctatatcatggccgacaagcagaagaacggcatcaaggtgaacttcaagatccgccacaacatcgaggacggcagcgtgcagctcgccgacactaccagcagaacacccccatcggcgacggccccgtgctgctgcccgacaaccactacctgagcacccagtccgccctgagcaaagaccccaacgagaagcgcgatcacatggtcctgctggagttcgtgaccgccgccgggatcactctcggcatggacgagctgtacaagAAGCGTCCTGCTGCTACTAAGAAAGCTGGTCAAGCTAAGAAAAAGAAAtagcagtatcgcaacgttgctgccgccgtggcagcagcagcagcggccgctgtcctcttcgtgtaagtatccacattgttctacaagtttcaatatatgtgtatattttacactttagagaacattttccatcatatatttctaacaataaactaacaaatctgttttaagattcaagatataatgaataaagaataacgttctcacttccttacgttccttaccacttacatatgtattgaaatcgaggaacagatcgacatt

Yellow—Homology arms

Turquoise—3X FLAG

Pink—T2A

Green—EGFP

Red—NLS, followed by Male stop TAG

Blue—gRNA

## Antibodies

The antibodies and RRIDs used in this study are listed in S1 Table. To generate Rabbit anti Zld-N, we used Pocono Rabbit Farm. Rabbits were injected with recombinant Zelda amino acids 1–618, which includes zinc fingers 1 and 2. The Zld-N antibody was pre-absorbed on $zld^{M-Z-}$ embryos prior to use.

## Dissection and immunofluorescence of adult gonads

Dissections of adult testes and ovaries were carried out as previously described [87]. Briefly, testes/ovaries were dissected in 1× phosphate-buffered saline (PBS), fixed for 30 min in 4% paraformaldehyde (PFA) in 1× PBS (testes). The ovaries were fixed for 30 min in 1× PBS with 4% PFA and 0.2% Triton X-100. The samples were washed two times for 30 min each at 25°C in 1× PBS with 0.5% Triton X-100, and blocked in PBTB (1× PBS, 0.2% Triton X-100 and 1% bovine serum albumin (BSA)) for 1 hour at 25°C. Primary antibodies were incubated overnight at 4°C. The samples were washed two times for 30 min each in 1× PBS with 0.2% Triton X-100 and incubated for 2 hours in secondary antibody in PBTB at 25°C and then washed two times for 30 min each in 1× PBS with 0.2% Triton X-100. They were mounted in Vectashield or Vectashield with DAPI (Vector Laboratories). Confocal images were captured using Zeiss LSM 700 and LSM 800 microscopes with a 63× objective.

For EcR detection with Ag10.2, WT ovaries, WT testes, *chinmo*-mutant testes, and Zld-overexpressing testes were dissected on the same day, stained with the same master mix of primary and secondary antibodies, and imaged on a confocal microscope using the same settings.

## Dissection and immunofluorescence of larval tissues

**Wing imaginal disc and ventral nerve cord.** For the dissection of larval wing imaginal discs and ventral nerve cords, properly staged larvae were rinsed in 1× PBS for immunofluorescence. The posterior part of the cuticle was carefully removed using forceps, and the specimens were inverted. The intestines and fat body were gently removed while keeping other organs intact and attached to the cuticle. Imaginal discs were dissected in 1× PBS and fixed for 30 min in 1× PBS containing 4% PFA and 0.2% Triton X-100 on a nutator. Following fixation, samples were washed three times for 10 min each in 1× PBS with 0.2% Triton X-100 and then blocked in PBTB (1× PBS with 0.2% Triton X-100 and 1% BSA) for 1 hour at room temperature. Primary antibodies were diluted in PBTB and incubated overnight at 4°C. After three 10-min washes in 1× PBS with 0.2% Triton X-100, samples were incubated with secondary antibodies in PBTB for 2 hours at room temperature. The samples were then washed three times for 10 min each in 1× PBS with 0.2% Triton X-100 before mounting in Vectashield or Vectashield with DAPI (Vector Laboratories). Confocal images were acquired using Zeiss LSM 700 and LSM 800 microscopes with a 40× objective.

**Larval gonads.** Male and female larvae were sexed based on gonad morphology. Male testes appeared as large, clear ovals embedded in the posterior third of the fat body, whereas female ovaries were smaller, clear, round spheres located in the same region. Larvae were dissected in 1× PBS, with the fat body and gonads carefully teased out while keeping the fat body attached to the larval cuticle. Other tissues, including the intestine, were removed.

The fat body with gonads was fixed in 4% PFA and 0.3% Triton X-100 in 1× PBS for 30 min with gentle rotation. Samples were washed twice for 10 min each in 1% PBST (1× PBS + 1% Triton X-100) and blocked in PBTB (1% PBST + 5% BSA) for 2 hours at room temperature. Primary antibodies were diluted in PBTB and incubated overnight at 4°C. Following three 20-min washes in 0.3% PBST at room temperature, samples were incubated with secondary antibodies in PBTB for 2 hours at room temperature. After a final 20-min wash in 0.3% PBST, samples were mounted in Vectashield or Vectashield with DAPI (Vector Laboratories).

### Hybridization Chain Reaction-Fluorescent in situ Hybridization (HCR-FISH)

All steps were done using RNase-free reagents and supplies with gentle rotation. The protocol for immunostaining with HCR-FISH was adapted from [88,89]. The HCR probe set against *zld*, *qkr58E-2*, and *chinmo* were purchased from Molecular Instruments, Briefly, testes/ovaries were fixed in 4% PFA in 0.1% Triton X-100 in 1×PBS-DEPC for 30 min at 25°C, washed with 0.5% Triton X-100 in 1× PBS-DEPC two times for 30 min at 25°C. Samples were blocked in 0.1% Triton X-100 in 1× PBS-DEPC with 50 μg/mL heparin and 250 μg/mL yeast tRNA (buffer hereafter called "PBTH"), and then they were incubated with primary antibodies overnight at 4°C. The next day, the samples were washed twice in PBTH for 30 min. Samples were then incubated with fluorescently labeled secondary antibodies in PBTH for 2 hours at 25°C. Samples were washed in PBTH twice for 30 min at 25°C. Samples were then dehydrated and rehydrated with a series of ethanol washes (25%, 50%, 75%, 100%) in 1× PBS-DEPC for 10 min at 25°C. Samples were treated for 7 min with 50 μg/mL Proteinase K, which was then inactivated by washing with 0.2% glycine twice in 1× PBS-DEPC for 5 min at 25°C. After Proteinase K treatment, the samples were fixed again in 4% PFA in 1× PBS-DEPC for 30 min at 25°C. The re-fixed samples were pre-hybridized in hybridization buffer provided by Molecular Instruments for 10 min at 25°C and then incubated with HCR probes overnight (12–16 hours) at 37°C. Samples were then washed 6 times 10 min at 37°C with wash buffer provided by Molecular Instruments and then twice for 5 min in 5× SCC at 25°C. Samples were incubated in amplification buffer provided by Molecular Instruments for 5 min at 25°C. The secondary reagents called "Hairpin h1 DNA" and "Hairpin h2 DNA" were prepared by heating each for 90 s at 95°C and cooling them at 25°C in a dark drawer for 30 min. Hairpin h1 DNA and Hairpin h2 DNA were mixed together at a 1:1 ratio and then added to the samples, which were then incubated in the dark environment overnight (16 hours) at 25°C. Samples were washed 6 times for 5 min with 5× SSC at 25°C and mounted in Vectashield plus DAPI for confocal analysis.

### Quantification of Zld protein, *zld* mRNA, and *qkr58E-2* mRNA

For Zld and GFP intensity quantifications, control and experimental samples were dissected and processed in parallel on the same day to minimize variability. Images were acquired using identical microscope settings. Regions of interest (ROIs) were drawn around somatic nuclei, identified by Tj or Zfh-1 signal, and mean fluorescence intensity was measured using the freehand selection tool in ImageJ. Background signal was subtracted using a defined background ROI. Appropriate thresholding was applied to ensure accurate quantification of fluorescence intensity. Data are presented as relative fluorescence intensity, normalized to the average control values.

For *zld* and *qkr58E-2* mRNA quantification, ROIs were similarly drawn around somatic nuclei based on Traffic jam (Tj) or Zfh-1 expression, and mean fluorescence intensity for *zld* and *qkr58E-2* was measured separately in ImageJ.

### miRNA in situ probes and hairpin amplifiers

DNA oligonucleotides were obtained from Integrated DNA Technologies, with sequences listed in S3 Table. DNA oligonucleotide probes were designed to contain a 20–24 nt sequence complementary to the target miRNA, flanked by initiators and linked by base linkers. Probe detection was carried out using HCR with 41-nt hairpin amplifiers labeled with Cy3. Hairpin amplifiers (Set B2, conjugated to Alexa-546) were purchased from Molecular Instruments

### miRNA HCR-FISH

All steps were done using RNase-free reagents and supplies with gentle rotation. The protocol for single-molecule FISH was adapted from [90]. Testes were dissected in 1× PBS and fixed in 4% formaldehyde in 1× PBS for 20 min. Testes were then washed twice for 5 min each in 1× PBS with 0.1% Tween 20 followed by permeabilization in 1× PBS with 0.1% Triton X-100 for 2 hours. Next, the testes were washed twice with 5× SSCT (0.1% Tween 20) for 5 min each. Probes were added to pre-warmed Probe Hybridization Buffer (Molecular Instruments) to a final concentration of 1 μM. A total of 100 μL of hybridization solution was added to each sample, pipetted to mix, and allowed to hybridize overnight at 37°C. Samples were then washed four times with Probe Wash Buffer (Molecular Instruments) for 15 min each at 37°C. During the first wash, amplifier hairpins (Molecular Instruments) were individually incubated at 95°C for 2 min and allowed to slowly cool to room temperature. Hairpins were then added to Amplification Buffer (Molecular Instruments) to a concentration of 60 nM. After washing the samples twice with 5× SSCT for 5 min each, 100 μL of hairpin solution was added and the samples were allowed to incubate at room temperature overnight. The testes were then washed twice with 5× SSCT for 30 min each at room temperature, mounted in Vectashield with DAPI, and imaged on Zeiss LSM 800 microscope with a 63× objective and processed using ImageJ and Adobe Photoshop software. WT, somatic *chinmo* depletion, and *miR* mutant (or miR sponge) samples were performed in parallel using a master mix of reagents and were imaged in parallel using the same confocal settings on the same day.

### Image analysis

Confocal images were acquired using Zeiss LSM 700 and LSM 800 microscopes with 40× or 63× objectives. Figures were exported from Zen (Zeiss software) and processed in Adobe Photoshop. Figures were prepared in Adobe Illustrator.

### Quantification of farthest Zfh-1-expressing CySC

Confocal z-stacks were acquired at 1 μm intervals to encompass the entire testis. Measurements were conducted using ImageJ on a single z-section, selected based on the clear visualization of both the niche (marked by Fas3) and a CySC (Zfh-1-positive, Eya-negative cell). To determine the distance of the farthest Zfh-1-positive, Eya-negative cell, a freehand straight line was drawn from the edge of the niche to the center of the most distal Zfh-1-expressing CySC.

### Raw data

Individual numerical values underlying all figures are in S1 Data

### Statistical analysis

Data were analyzed and graphs were generated using GraphPad Prism software. For comparisons between two groups, unpaired two-tailed Student *t*-tests were performed. Datasets based on individual measurements are presented as scatter plots showing all data points. Categorical outcomes are displayed as stacked bar graphs, with each bar representing 100% of the samples for a given genotype. The colored segments within each bar indicate the proportion of samples in each category (e.g., feminized versus non-feminized, shown in pink and blue, respectively). Two-tailed Fisher's exact tests were used for the analysis of categorical data. The specific statistical tests, sample sizes, *P*-values, and significance indicators are provided in the corresponding figure legends, and all data points were included in the analyses.

### Supporting information

**S1 Fig. Schematic of WT testis, WT ovary and feminized testis. (A)** Cyst stem cells (CySCs, dark blue) of a wild-type (WT) testis are mitotic and reside in the same niche (green) as germline stem cells (GSCs, gray). Their daughter cells (cyst cells, light blue) are squamous, quiescent, and encyst the differentiating male germ cells (gray). **(B)** Follicle stem

cells (FSC, dark red) reside in the germaria of a WT ovary. They give rise to follicle cells (FC, light red) that are mitotic and epithelial. FCs form an epithelium that encyst differentiating female germ cells (gray). The ovary also has a niche (green) that supports GSCs (gray). Escort cells (ECs, pink) envelope early female germ cells. **(C)** In a testis somatically depleted of *chinmo*, the niche (green) remains and the CySCs transdifferentiate into FSC-like cells (dark pink). These FSC-like cells give rise to FC-like cells (pink) that cannot properly support male germ cells, leading to defective spermatogenesis and infertility.
(TIF)

**S2 Fig. Gene ontology for biological processes enriched in genes from *chinmo*-deficient CySCs with predicted Zld binding sites.** Representative enrichment of upregulated biological processes using Database for Annotation, Visualization and Integrated Discovery (DAVID) classification database. The data underlying the graph shown in the figure can be found in S1 Data.
(TIF)

**S3 Fig. Validation of Zld antibody, Zld-tagged alleles, and UAS-*zld*-RNAi line. (A, B)** Representative confocal images of control (*dpp>+*) **(A)** and *dpp > zld-i* **(B)** wing imaginal discs stained for Zld (red, grayscale), Dpp domain (green), and DAPI (blue). Yellow dotted lines **(B″)** show the diminished expression of Zld in *dpp > zld-i*. **(C–F)** Representative confocal images from control testes (*tj>+*) displaying negligible Zld expression **(C)** or faint Zld expression **(D)**, from *tj > zld-i* testes **(E)**, and *tj > zld* testes **(F)**. Testes are stained for Fas3 (grayscale), Zfh-1 (magenta, grayscale), and Zld (green, grayscale). **(G)** Graph showing the percentage of testes displaying faint (gray) and negligible (black) Zld protein expression in *tj>+* ($n=36$) and *tj>zld-i* ($n=25$). **(H)** Graph showing relative Zld protein expression in somatic cells of *tj>+* ($n=12$) and *tj>zld* ($n=12$) testes. **(I)** Graph depicting the total number of Zfh-1-positive cells in *tj>+* ($n=11$) and *tj>zld-i* ($n=11$) testes. **(J–L)** Representative confocal images of sfGFP-Zld in the larval ventral nerve cord (VNC) **(J)**, wing imaginal disc **(K)**, and adult testis **(L)**. sfGFP-Zld is shown in green and grayscale. DAPI is marked in red. **(M)** Graph showing the percentage of feminized (pink) and non-feminized (blue) testes in *FM7/Y; chinmo^{ST}/CyO* ($n=15$), *FM7/Y; chinmo^{ST/ST}* ($n=20$), and *sfGFP-Zld/Y; chinmo^{ST/ST}* ($n=15$). **(N–P)** Representative confocal images of mNG-Zld in larval VNC **(N)**, wing imaginal disc **(O)**, and adult testis **(P)**. mNG-Zld is shown in green and grayscale. DAPI is marked in red. **(Q)** Graph showing the percentage of feminized (pink) and non-feminized (blue) testes in *FM7/Y; tj>+* ($n=15$), *FM7/Y; tj>chinmo-i* ($n=20$), and *mNG-Zld/Y; tj>chinmo-i* ($n=20$). Dot plots show individual data points with lines indicating the mean $\pm$ SD **(H, I)**. Bar graphs depict the percentage of testes exhibiting the indicated phenotypes **(G, M, Q)**. The data underlying the graphs shown in the figure can be found in S1 Data. Statistical analysis was performed using Student *t* test **(H, I)** and Fisher's exact test **(G, M, Q)** (ns = not significant; ** $P = 0.0078$; **** $P < 0.0001$). Scale bars: 20 μm in **A**–**F**, **L**, **P**, and 50 μm in **J**, **K**, **N**, **O**.
(TIF)

**S4 Fig. Zld is not expressed in adult or larval ovary and is not required in oogenesis. (A)** Representative confocal images of a WT adult ovary stained for Tj (blue, grayscale), Fas3 (red, grayscale), and Zld (green, grayscale). **(B–D)** Representative confocal images of WT larval ovary **(B)**, WT larval testis **(C)**, and WT larval VNC **(D)** stained for Tj (red, grayscale), Zld (green, grayscale), and Fas3 (blue, grayscale in **C**). DAPI (blue, grayscale in **B**, **D**). **(E)** Representative confocal images of *c587^{ts}>zld-i* ovary stained for Tj (blue, grayscale), Fas3 (green, grayscale), and Vasa (red, grayscale). Scale bars: 20 μm.
(TIF)

**S5 Fig. Validation of Dcr-1 RNAi lines and data showing that *miR-283* promotes differentiation of somatic cells in the testis. (A, B)** Representative confocal images of *c587>+* (A) and *c587>Dcr-1-i* (B) testes stained for Dcr-1 (green, grayscale), Fas3 (red, grayscale), and Tj (blue, grayscale). **(C)** Graph showing relative Dcr-1 expression in somatic cells of *c587>+* ($n=15$), *c587>Dcr-1-i* #1 ($n=15$), and *c587>Dcr-1-i* #2 ($n=9$) testes. **(D, E)** Representative confocal images

of WT (D) and *miR-283KO* (E) testes stained for Vasa (grayscale), Zfh-1 (magenta, grayscale), and Fas3 and Eya (green, grayscale). **(F)** Graph depicting the total number of Zfh-1-positive, Eya-negative cells in WT ($n = 9$) and *miR-283KO* ($n = 11$) testes. **(G)** Graph showing distance of the farthest Zfh-1-positive, Eya-negative cells in WT ($n = 9$) and *miR-283KO* ($n = 11$) testes. Dot plots show individual data points with lines indicating the mean ± SD **(C, F, G)**. The data underlying the graphs shown in the figure can be found in S1 Data. Statistical analysis was performed using Student *t* test **(C, F, G)** (*** $P = 0.0004$; **** $P < 0.0001$). Scale bars: 20 μm.
(TIF)

**S6 Fig. Ir93a and Gmap depletion have no effect on the CySC lineage. (A–C)** Representative confocal images of *tj*>+ **(A)**, *tj* > *Ir93a-i* **(B)**, and *tj*>*Gmap-i* **(C)** testes stained for Zld (green, grayscale), Zfh-1 (blue, grayscale), and Fas3 (red). A‴, B‴, and C‴ show Z-max projections of Zfh-1-expressing cells. **(D)** Graph depicting the total number of Zfh-1-positive cells in *tj*>+ ($n = 11$), *tj* > *Ir93a-i* ($n = 11$), and *tj*>*Gmap-i* ($n = 12$) testes. Dot plots show individual data points with lines indicating the mean ± SD **(D)**. The data underlying the graphs shown in the figure can be found in S1 Data. Statistical analysis was performed using Student *t* test **(D)** (ns = not significant). Scale bars: 20 μm.
(TIF)

**S7 Fig. Validation of *qkr58E-2* RNAi and *qkr58E-2^{G3095}* EP lines. (A, B)** Representative confocal images of HCR-FISH for *qkr58E-2* mRNA in *tj*>+ **(A)** and *tj* > *qkr58E-2-i* **(B)** ovaries. Ovaries are stained for Fas3 (green), Tj (blue), and *qkr58E-2* mRNA (grayscale, inverted grayscale). The panels **A″** and **B″** shows the enlarged view of the inset marked with yellow boxes in **A′** and **B′**. **(C)** Graph showing relative *qkr58E-2* mRNA intensity in somatic cells in *tj*>+ ($n = 8$), *tj* > *qkr58E-2-i* #1 ($n = 10$), and *tj* > *qkr58E-2-i* #2 ($n = 8$) ovaries. **(D, E)** Representative confocal images of HCR-FISH for *qkr58E-2* mRNA in *c587*>+ **(D)** and *c587* > *qkr58E-2^{G3095}* **(E)** testes. The testes are stained for Fas3 (green), Tj (blue), and *qkr58E-2* mRNA (grayscale, inverted grayscale). **(F)** Graph showing relative *qkr58E-2* mRNA intensity in somatic cells in *c587*>+ ($n = 8$) and *c587* > *qkr58E-2* ^{G3095} ($n = 9$) testes. Dot plots show individual data points with lines indicating the mean ± SD **(C, F)**. The data underlying the graphs shown in the figure can be found in S1 Data. Statistical analysis was performed using Student *t* test (C, F) (**** $P < 0.0001$). Scale bars: 20 μm.
(TIF)

**S8 Fig. *tra* alternative splicing and differentiation of ovarian follicle cells depend on *qkr58E-2*. (A, B)** Representative confocal images of *tj*>*traF^{ΔT2AGFP}* + **(A)** and *tj*>*traF^{ΔT2AGFP}; qkr58E-2-i* ovaries **(B)** stained for GFP (indicative of Tra^F) (yellow, inverted grayscale), Fas3 (green, inverted grayscale), and Tj (magenta, inverted grayscale). Blue and red brackets in **B″** and **B‴**, respectively, show reduced Fas3 and Tra^F in *qkr58E-2*-depleted follicle cells. **(C)** Graph showing the percentage ovaries with (pink) or without (gray) Tra^F in *c587*>*traF^{ΔT2AGFP}* + ($n = 65$), *c587*>*traF^{ΔT2AGFP}; qkr58E-2-i* #1 ($n = 98$), *c587*>*traF^{ΔT2AGFP}; qkr58E-2-i* #2 ($n = 71$), *tj*>*traF^{ΔT2AGFP}* + ($n = 55$), *tj*>*traF^{ΔT2AGFP}; qkr58E-2-i* #1 ($n = 68$), and *tj*>*traF^{ΔT2AGFP}; qkr58E-2-i* #2 ($n = 56$). **(D–F)** Representative confocal images of *tj*>+ **(D)**, *tj*>*qkr58E-2-i* **(E)**, and *tj^{ts}*>*dsx-i* **(F)** ovaries stained for Dcp-1 (yellow, inverted grayscale), Fas3 (green, inverted grayscale), Tj (magenta, inverted grayscale), and DAPI (inverted grayscale). Green brackets in **E″** and **F″** show fused egg chambers and abnormal germaria in ovaries somatically depleted for *qkr58E-2* **(E″)** and *dsx* **(F″)**. Brown arrows in **E‴** and **F‴** show Dcp-1 expression in follicle cells depleted for *qkr58E-2* **(E‴)** and *dsx* **(F‴)**. **(G)** Graph showing defective oogenesis and cell death in *tj*>+ ($n = 72$), *tj*>*qkr58E-2-i* #1 ($n = 52$), *tj*>*qkr58E-2-i* #2 ($n = 37$), *tj^{ts}*>*dsx-i* #1 ($n = 58$), and *tj^{ts}*>*dsx-i* #2 ($n = 32$) ovaries. Bar graphs depict the percentage of ovaries exhibiting the indicated phenotypes **(C, G)**. The data underlying the graphs shown in the figure can be found in S1 Data. Statistical analysis was performed using Fisher's exact test **(C, G)** (**** $P < 0.0001$). Scale bars: 20 μm.
(TIF)

**S9 Fig. *qkr58E-2* depletion does not impact the CySC lineage in the testis. (A, B)** Representative confocal images of *tj*>+ **(A)** and *tj* > *qkr58E-2-i* **(B)** testes. The testes are stained for Fas3 (blue), Tj (red, grayscale), and Zfh-1 (green,

grayscale). **A‴** and **B‴** show Z-max projections of Zfh-1-expressing cells. **(C)** Graph depicting the total number of Zfh-1-positive cells in *tj>+* (*n* = 10) and *tj > qkr58E-2-i* (*n* = 10) testes. Dot plots show individual data points with lines indicating the mean ± SD **(C)**. The data underlying the graphs shown in the figure can be found in S1 Data. Statistical analysis was performed using Student *t* test **(C)** (ns = not significan*t*). Scale bars: 20 μm.
(TIF)

**S10 Fig. Overexpression of Tra^F and Dsx^F in male somatic cells does not trigger loss of Chinmo or feminization.** **(A–C)** Representative confocal images of *c587^ts^>+* **(A)**, *c587^ts^>tra^F* **(B)**, and *c587^ts^>dsx^F* testes at 20 days of adulthood. The testes are stained for Fas3 (grayscale), Castor (green, grayscale), and Chinmo (magenta, grayscale). **(D)** Graph showing the percentage of feminized (pink) and non-feminized (blue) testes in *c587^ts^>+* (*n* = 10), *c587^ts^>tra^F* (*n* = 10), and *c587^ts^>dsx^F* (*n* = 10). Bar graphs depict the percentage of testes exhibiting the indicated phenotypes **(D)**. The data underlying the graphs shown in the figure can be found in S1 Data. Statistical analysis was performed using Fisher's exact test **(D)** (ns = not significant). Scale bars: 20 μm.
(TIF)

**S11 Fig. Schematic for *Dsx^M^::GFP allele*.** See Materials and methods for detailed information.
(TIF)

**S1 Table. List of key reagents and resources used in this study, including primary and secondary antibodies, chemicals, and *Drosophila* strains.** All *Drosophila* stocks are provided with their corresponding RRIDs.
(DOCX)

**S2 Table. List of all *Drosophila* genotypes examined in the main figures and supporting information.**
(DOCX)

**S3 Table. All probes and oligonucleotides used for HCR-FISH are detailed here, including the HCR probe sets for *qkr58E-2, zld,* and *chinmo*.** The table additionally includes miRNA initiator probe sequences, target miRNAs, amplifier sets, probe lengths, and associated DNA oligonucleotide sequences.
(DOCX)

**S1 Data. All individual data values corresponding to the graphical representations in the main figures and supporting information.**
(XLSX)

## Acknowledgments

We are indebted to Amelie Raz (Whitehead Institute, MIT, USA) for providing essential information about generating probes to detect miRNAs. We thank K. White, P. Zamore, D. Godt, P. Rangan, N. Sokol, and DHSB for antibodies, and E. Matunis, R. Lehmann, M. Mir, M. Harrison, B. Oliver, and BDSC for fly stocks.

## Author contributions

**Conceptualization:** Sneh Harsh, Erika A. Bach.

**Data curation:** Sneh Harsh.

**Formal analysis:** Sneh Harsh.

**Funding acquisition:** Christine Rushlow, Erika A. Bach.

**Investigation:** Sneh Harsh.

**Methodology:** Sneh Harsh, Erika A. Bach.

**Project administration:** Erika A. Bach.

**Resources:** Hsiao-Yun Liu, Pradeep K Bhaskar, Christine Rushlow, Erika A. Bach.

**Supervision:** Erika A. Bach.

**Validation:** Sneh Harsh.

**Visualization:** Sneh Harsh.

**Writing – original draft:** Sneh Harsh, Erika A. Bach.

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
