## [Editor Report · Decision Letter 0]

20 Apr 2025

Dear Dr Bach,

Thank you for submitting your manuscript entitled "Post-transcriptional suppression of the pioneer factor Zelda protects the adult testis from activation of the ovary program" for consideration as a Research Article by PLOS Biology.

Your manuscript has now been evaluated by the PLOS Biology editorial staff as well as by an academic editor with relevant expertise and I am writing to let you know that we would like to send your submission out for external peer review.

Once your full submission is complete, your paper will undergo a series of checks in preparation for peer review. After your manuscript has passed the checks it will be sent out for review. To provide the metadata for your submission, please Login to Editorial Manager (https://www.editorialmanager.com/pbiology) within two working days, i.e. by Apr 22 2025 11:59PM.

Kind regards,

Ines

--

Ines Alvarez-Garcia, PhD

Senior Editor

PLOS Biology

---

## [Decision Letter · Decision Letter 1]

25 Jun 2025

Dear Dr Bach,

Thank you for your patience while your manuscript entitled "Post-transcriptional suppression of the pioneer factor Zelda protects the adult testis from activation of the ovary program" was peer-reviewed at PLOS Biology. Please also accept again my apologies for the delay sending you our decision. The manuscript has now been evaluated by the PLOS Biology editors, an Academic Editor with relevant expertise, and by three independent reviewers.

The reviews are attached below. As you will see, the reviewers find the conclusions interesting and novel, however they have also raised several issues that would need to be addressed before we can consider the manuscript for publication. Reviewer 1 asks for the clarification of several points and also proposes an alternative explanation for some of the findings that should be considered. Reviewer 2 suggests further analyses to strengthen the data, including mining available multiomic data from the Fly Cell Atlas to determine if specific TAGteam sites change in their accessibility status in different tissues or in males vs females, characterising further these genes in GO and measuring Zelda RNA levels to confirm that the transcript is regulated by the miRNAs identified. Reviewer 3 thinks it should be tested if Zelda is a very sensitive target of miRNAs given that multiple miRs have an effect either on expression of Zelda or in testis cytology. This reviewer also notes that there are no combinations of miRNA mutants shown in Fig. 3 and that some of them should be tested, as they might have strong effects.

The Academic Editor has added further comments on the reviews as following:

Reviewer 1:

Both comments on the Introduction about ovary and taxa would be easy to address with text-revision, and important. Comment 1 on Results about how loss of chinmo results in feminizing the testis, whereas loss of miRNA does not. Reviewer 2 has the same concern and it would be important to address it. On the discussion, the authors should consider the first paragraph and ideally address it in the manuscript. While they should consider the interesting speculation as well, they can decide whether to address it in the rebuttal or with changes to the text.

Reviewer 2:

The first two comments about TAGteam sites are easy to address with available literature and appropriate revisions to the text. The Gene Ontology request would be an excellent addition, but it would understandable if the authors decide not to include it. In addition, the Zelda RNA level experiment would be nice to have, but it is uncertain if the results would distinguish among the possibilities that the reviewer raises, so it should be optional to the authors. The last comment is similar to the Results comment from Reviewer 1 and should be addressed in the revised manuscript. Minor comments should be up to the authors' discretion.

Reviewer 3:

The first comment is interesting and if it is easy to test it would be nice if the authors can address it. Otherwise, it should be addressed in writing. The second comment doesn’t seem easy and we won’t insist on it. Minor comment #1 is not something that needs to be added to the paper and for Minor comment #2, the authors should make the word-change in the text.

In light of the reviews, we would like to invite you to revise the work to thoroughly address the reviewers' reports in light of the Academic Editors' comments. Given the extent of revision needed, we cannot make a decision about publication until we have seen the revised manuscript and your response to the reviewers' comments. Your revised manuscript is likely to be sent for further evaluation by all or a subset of the reviewers.

**IMPORTANT - SUBMITTING YOUR REVISION**

3. Resubmission Checklist

a) *PLOS Data Policy*

b) *Published Peer Review*

Sincerely,

Ines

--

Ines Alvarez-Garcia, PhD

Senior Editor

PLOS Biology

Reviewers' comments

Rev. 1:

In general, I found the paper very informative and well written. It fills an important gap in our understanding of how sex reversal can occur in fully differentiated adult organs, so it should be of interest to a wide audience. The experiments are carefully conducted and well described. The use of multiple experimental approaches and well-chosen controls gives one strong confidence in the results. The writing is straightforward, and the figures are clear and well organized. I also like how the data are divided between the main text and the supplement. All in all, I think this is an excellent manuscript and I only have a few comments.

Introduction

The authors never explain that chinmo is not expressed in the normal ovary. I understand that this background is well established, but without this piece of data, the logic doesn't make any sense, so it's worth saying explicitly.

This is nitpicking, but in Lines 64-69, please use more careful phrasing. Mammals and flies are not "species"; and there are flies (such as Musca and Calliphora) that do not require a match between somatic and germline sex.

Results

There's one set of results that I don't know what to make of. One of the central ideas in this paper is that (1) increased abundance of the Zld protein is the key mechanism by which the loss of chinmo induces testis feminization and (2) the effect of chinmo on Zld levels is mediated by the miRNAs miR283, miR1011, and miR263a. However, loss of chinmo results in the feminization of the testis, while miRNA loss does not (lines 205-206). This is despite the fact that, as far as I can tell, the changes in Zld levels are comparable. Zld expression appears to be increased roughly 3.5-4.5x in chinmo mutants and RNAi KO, 2.5x in miR283 KO, 6x in miR1011 KO, and 4-5x in the miR263a sponge (Fig 2). How do the authors explain this discrepancy? Do they believe that the increase in Zld levels that occurs following single miRNA loss is not as strong as what's seen in chinmo KO? Or is there some parallel mechanism through which chinmo acts independently of Zld? An obvious experiment would be to test double and triple miRNA mutants, though I imagine this experiment is technically difficult. Did the authors attempt it? A related observation is that a ~3-3.5x increase in Zld levels is seen in Dcr-1 KO (Supplement Fig 4). Does that result in testis feminization? The authors' model would seem to predict that it should, but I could not find any data on that. Anyway, some discussion of the discrepancies between the chinmo and miRNA phenotypes, despite the fact that both derepress Zld, would be welcome.

Discussion

Prolonged Zld expression reduces chinmo transcription. EcR is higher in the ovary than testis, and is upregulated in chinmo mutants. Ectopic Zld causes higher EcR expression. An obvious hypothesis is that EcR mediates the long-term effect of Zld on chinmo expression. As the authors say in the Discussion section, the EcR story is complicated, and some results are contradictory. Nevertheless, it would be nice to discuss the evidence for and against this hypothesis directly, since it could have substantial explanatory power. At the moment, that section of the Discussion is the least clear (to me, anyway).

My other point is more speculative. The authors present zld as having a somewhat specific function in female sexual differentiation. But another way to look at it is that zld, as a pioneer TF, may have a very generic role in transdifferentiation and cell fate reassignment (in addition to its "normal" roles in early development). In this case, zld is not expressed or required for normal ovary development in XX flies - it only seems to be involved in transdifferentiation of testis into ovary in mutant XY flies. In another context, zld is dispensable for normal wing development, but is required for cell fate respecification during wing disc regeneration (https://pubmed.ncbi.nlm.nih.gov/38854062/). It can also reprogram partially differentiated neuroblasts back toward a neural stem cell identity (https://pubmed.ncbi.nlm.nih.gov/34887421/). So, is it possible that zld does not induce sex reversal, but rather permits cell fate reprogramming more generally by allowing reactivation of previously silenced genes? In this case, there are almost 100 genes that are downstream of chinmo and have Zld binding sites, but the effect of chinmo on the splicing of tra (and thus feminization) seems to be largely facilitated by just one, qkr58E-2. Is it possible that ectopic expression of Zld simply opens up a random subset of its target genes, and that the instructive role in sex reversal is played by a different upstream regulator of qkr58E-2? Again, this is pure speculation. I am not suggesting that this alternative interpretation is correct, and I am not asking for an experimental test of this idea. I'm just bringing it up as a possibility. And perhaps it's worth considering zld functions in other contexts in the Discussion?

Rev. 2:

Review: Post-transcriptional suppression of the pioneer factor Zelda protects the adult testis from activation of the ovary program

Summary:

Bach and colleagues seek to investigate the role of Zelda in somatic sex identity of tests and sex reversal of testes. The male identity of the adult somatic stem cells (cyst cells) in the testis requires the transcription factor Chinmo and through loss of chinmo, the testes are feminized and acquire ovary programming. Here the authors investigated the differentially expressed genes in chinmo depleted testis and found that 30% of the genes contained Zelda binding sites. The authors showed that in WT testes, the Zelda protein is not present via confocal imaging. To further investigate the role of Zelda in male identity of the testes cyst cells, the author depleted both zelda and chinmo in the cyst cells. By Fas3 assay, the testes in chinmo depleted cells were feminized while there was significantly less feminization in the cells where they depleted both zelda and chinmo. The authors then discovered that the Zelda transcript levels in chinmo mutants were unchanged despite the fact that there is more Zelda protein. They investigated further to interrogate whether zld transcripts are repressed by miRNAs in WT cyst cells. Using candidate miRNAs they hypothesized that chinmo was responsible for expressing the miRNAs that regulated zld mRNA repression. They found three miRNAs that were reduced in the chinmo mutant background, and also corresponded to an increase in Zelda protein when each of the three miRNAs were mutated. However, testes were not feminized in the single miRNA mutants so there is some redundancy. To test whether Zelda is sufficient to trigger feminization, the authors monitored the alternative splicing product TraF only present in WT ovarian FCs. The authors showed that ectopic expression of zld in cyst cells induced TraF splicing however this did not reach levels in WT ovarian FCs. The authors next looked at how Zelda might be impacting the splicing of TraF and found that in chinmo mutants, transcript levels of an RBP (Qkr58E-2) was misregulated.

This is a strong manuscript with high quality immunostaining and quantification. It addresses an important question of how gonads maintain sexual differentiation and avoid inappropriately switching sexes. The authors identified a novel microRNA pathway that is used to repress the induction of Zelda and used both antibody staining and tagged Zelda as tools to address their hypothesis.

Questions:

The authors note that 30% of the target genes they identify have TAGteam sites and therefore might be regulated by Zelda. However, the presence of a binding site alone is not highly predictive of binding.

Can the authors mine available multi-omic data from the Fly Cell Atlas to determine if these specific TAGteam sites change in their accessibility status in different tissues or in males versus females?

Additional characterization of these genes in terms of Gene ontology analysis would be helpful.

Can the authors measure zelda RNA levels by qPCR or FISH to show that the transcript is being regulated by the microRNAs identified because the current studies rely only on immunostaining? Therefore, it is possible that the microRNAs are functioning indirectly through targeting of another factor that regulates Zelda protein levels.

The Qkr58E-2 experiments are a little counterintuitive: Even with more Zelda, there is no feminization without the RBP, so wouldn't Qrk58E-2 be more important for maintaining the "male identity" and preventing female identity of the somatic cells.

Zelda intensity is increased to the same amount in the miRNA mutant as in Figure 1 BUT there is no feminization. How do the authors reconcile this?

Minor Comments on text:

Ending of the introduction jumps around a little, hard to follow logic after mentioning the targets of Zelda in the embryo

Why do RNAi and the KO

Miss expressing vs overexpressing is confusing

The phrase "Zelda is required for conversion of adult male gonadal somatic cells into their female counterparts" is confusing:

Because feminizing is NOT a normal phenomenon it would be good to rephrase: Zelda is important for maintaining MALE somatic stem cell identity and preventing feminization?

Move "sustained ectopic Zld suppresses…" paragraph after the RBP stuff and right before the "ectopic EcR expression…" paragraph

The model figure is confusing with different parts greyed out. It would be better to use an arrow with a line at the end to show repression.

Rev. 3:

In their study "Post-transcriptional suppression of the pioneer factor Zelda protects the adult testis from activation of the ovary program", Harsh and Bach et al use genetic strategies to elucidate a multistep regulatory program that controls sexual identity.

Their stated highlights are:

* zld mRNA is repressed by microRNAs in XY somatic gonadal cells

* Zld is upregulated in and required for sex reversal of XY chinmo-/- cells

* Zld induces Qkr58E-2 and EcR, which cause TraF and female-biased transcription

* Zld feminizes XY adipose cells by inducing TraF and downregulating Chinmo

I think these conclusions supported by their data and are both interesting and thorough. In particular, the imaging data are compelling and beautiful, and new regulatory links are uncovered. On the miRNA side, they used both mutant (knockout or inhibitor) and misexpression studies to demonstrate phenotypic impacts of several miRNAs to suppress zelda in testis soma. One note is that it is fast becoming a standard to engineer mutations of miRNA binding sites as definitive proof of the biological impact of miRNA mediated regulation on specific target genes. This is due to the fact that miRNAs tend to have many targets, which might or might contribute to the phenotype of the miRNA mutant. However, since this study goes beyond simply focusing on zelda as a target of miRNAs, I think that it can be left to future studies to generate mutations of miRNA binding sites in zelda.

Overall, I think they did a great job to elaborate a pathway upstream of zelda (miRNAs) and downstream of zelda (chinmo and Qkr58E-2), thus involving transcriptional and post-transcriptional effectors. I have only some small comments that I think would be easy to do.

Major comments

1. Its very notable that multiple miRNAs had an effect either on expression of Zelda, or in testis cytology. This is interesting because many if not most miRNAs have relatively little effects, and if there are multiple functionally overlapping miRNAs, they might compensate for each other. This could be interpreted that Zelda is a very sensitive target of miRNAs. Given their in situs indicate co-expression of the miRNAs, this is a very testable scenario.

I might have missed if they did it, apologies if so, but I noted Fig 3 had single miRNA mutants, and there are tests with multiple miRNA-GOF, but not combinations of miRNA mutants. I think there is a strong expectation that combination miRNA mutants would yield strong effects. Since they have the reagents, I think its not hard to test at least some combinations.

Minor comments

1. I am curious about the miRNA in situs, since this is known to be a difficult technique. My understanding is that one can always get some signals by developing the reaction longer or by increasing gain etc. I am wondering how they determine that some miRNA in situ probes don't provide evidence of signal. This may be worth documenting in the supplement, and maybe comparing exposure times with real miRNA signals.

2. "Figure 2: zld is post-transcriptionally modified"

It is more appropriate to state that zelda is post-transcriptional regulated, rather than modified, as the latter implies the RNA is subject to an epitranscriptomic modification.

---

## [Decision Letter · Decision Letter 2]

19 Oct 2025

Dear Dr Bach,

Thank you for your patience while we considered your revised manuscript entitled "Post-transcriptional suppression of the pioneer factor Zelda protects the adult Drosophila testis from activation of the ovary program" for publication as a Research Article at PLOS Biology. This revised version of your manuscript has been evaluated by the PLOS Biology editors, the Academic Editor and one of the original reviewers.

Based on the reviews, we are likely to accept this manuscript for publication, provided you satisfactorily address the remaining data and other policy-related requests stated below my signature.

We expect to receive your revised manuscript within two weeks.

*Published Peer Review History*

*Press*

Sincerely,

Ines

--

Ines Alvarez-Garcia, PhD

Senior Editor

PLOS Biology

DATA POLICY:

Many thanks for providing the data underlying the graphs shown in the figures. I have checked the data and I would like you to address the following points:

- The data underlying the graphs show in Fig. S2 seem to be missing - please provide it.

- Data from Fig. 6D seem to be mislabelled in the S1 Data file – please correct it.

- Fig. S5C seems to miss some data – please check it and add the data.

- Please include in all the corresponding figure legends where the underlying data can be found. For example, you can add at the end: "The data underlying the graphs shown in the figure can be found in S1 Data."

Reviewers' comments

Rev. 1:

I thank the authors for addressing all my questions, and for performing new experiments and adding new data that were needed to answer these questions. A couple of issues that remain unresolved are explicitly pointed out in the revised manuscript. I therefore have no further comments.

---

## [Editor Report · Decision Letter 3]

13 Nov 2025

Dear Dr Bach,

Thank you for the submission of your revised Research Article entitled "Post-transcriptional suppression of the pioneer factor Zelda protects the adult Drosophila testis from activation of the ovary program" for publication in PLOS Biology. On behalf of my colleagues and the Academic Editor, Mariana Wolfner, I am delighted to let you know that we can in principle accept your manuscript for publication, provided you address any remaining formatting and reporting issues. These will be detailed in an email you should receive within 2-3 business days from our colleagues in the journal operations team; no action is required from you until then. Please note that we will not be able to formally accept your manuscript and schedule it for publication until you have completed any requested changes.

PRESS

Sincerely, 

Ines

--

Ines Alvarez-Garcia, PhD

Senior Editor

PLOS Biology
